# Recurring seasonality exposes dominant species and niche partitioning strategies of open ocean picoeukaryotic algae
Charlotte A. Eckmann [1,2], Charles Bachy [3,4], Fabian Wittmers[1,3], Jan Strauss [3],
Leocadio Blanco-Bercial [5], Kevin L. Vergin[6], Rachel J. Parsons[5], Raphael M. Kudela[2], Rod Johnson[5],
Luis M. Bolaños [7], Stephen J. Giovannoni[7], Craig A. Carlson[8] & Alexandra Z. Worden [1,2,3] ✉

Ocean spring phytoplankton blooms are dynamic periods important to global primary production. We document vertical patterns of a diverse suite of eukaryotic algae, the prasinophytes, in the North Atlantic Subtropical Gyre with monthly sampling over four years at the Bermuda Atlantic Time-series Study site. Water column structure was used to delineate seasonal stability periods more ecologically relevant than seasons defined by calendar dates. During winter mixing, tiny prasinophytes dominated by Class II comprise 46 ± 24% of eukaryotic algal (plastid-derived) 16S rRNA V1-V2 amplicons, specifically *Ostreococcus* Clade OII, *Micromonas commoda*, and *Bathycoccus calidus*. In contrast, Class VII are rare and Classes I and VI peak during warm stratified periods when surface eukaryotic phytoplankton abundances are low. Seasonality underpins a reservoir of genetic diversity from multiple prasinophyte classes during warm periods that harbor ephemeral taxa. Persistent Class II sub-species dominating the winter/spring bloom period retreat to the deep chlorophyll maximum in summer, poised to seed the mixed layer upon winter convection, exposing a mechanism for initiating high abundances at bloom onset. Comparisons to tropical oceans reveal broad distributions of the dominant sub-species herein. This unparalleled window into temporal and spatial niche partitioning of picoeukaryotic primary producers demonstrates how key prasinophytes prevail in warm oceans.

Ocean subtropical gyres are important regions of net primary production (NPP) by phytoplankton and carbon export due to their areal expansiveness[1,2]. The North Atlantic Subtropic Gyre is currently considered a net $CO_2$ sink due to phytoplankton uptake of atmospheric $CO_2$, with an observed summertime efflux of $CO_2$ from the ocean to the atmosphere that is more than offset by $CO_2$ uptake during a winter/spring bloom[3]. This ecosystem has been well characterized at the Bermuda Atlantic Time-series Study (BATS) site where a prominent feature is strong seasonality that shapes water column structure, biogeochemical cycling, phytoplankton successional patterns, productivity, and export[4,5]. An annual deep convective mixing event (DM) between January and April results in the entrainment of nutrients from the mesopelagic zone into the euphotic zone that is fundamental to the seasonal and successional patterns observed.

Subsequent thermal stratification in the spring and summer allow for macronutrient draw-down without replenishment, resulting in low concentrations of nutrients in the surface waters during summer[6]. This seasonal stratification is considered analogous to current concepts of future desertification and expansion of oligotrophic regions of the ocean[7]—with varied possible consequences for primary production[8].

Ocean spring blooms are large and dynamic biological events[9]. At BATS, the highest NPP recorded seasonally has historically been during the winter/spring bloom period, coincident the greatest fluxes of particulate organic carbon (POC)[6,10,11]. In general, the phytoplankton responsible for the bulk of NPP in subtropical regions are picoplanktonic (plankton with ≤ 2 μm cell diameter); their small size results in a greater surface area to volume ratio and is advantageous when competing for scarce nutrients.

[1]Marine Biological Laboratory, Woods Hole, MA 02543, USA. [2]Ocean Sciences Department, University of California, Santa Cruz, CA 95064, USA. [3]Ocean EcoSystems Biology Research Unit, GEOMAR Helmholtz Centre for Ocean Research Kiel, Kiel 24148, Germany. [4]Station Biologique de Roscoff, Sorbonne Université, CNRS, FR2424, Roscoff 29680, France. [5]Bermuda Institute of Ocean Sciences—Arizona State University, St. George's GE 01, Bermuda. [6]Microbial DNA Analytics, Phoenix, OR 97535, USA. [7]Department of Microbiology, Oregon State University, Corvallis, OR 97331, USA. [8]Department of Ecology, Evolution, and Marine Biology, University of California, Santa Barbara, CA 93106, USA. ✉e-mail: azworden@mbl.edu

Phytoplankton distributional patterns at BATS have been characterized at the level of major groupings[6,12,13]. However, capturing a deep convective mixing event at the time nutrient entrainment occurs can be difficult due to the ephemeral nature of the events. While studies using representative pigments to identify phytoplankton groups have been instrumental in gaining a general sense of how different phytoplankton communities contribute to NPP, they do not resolve key open ocean groups[14,15]. Nevertheless, it is clear that the greatest eukaryotic contributions occur during the DM and bloom period, while cyanobacterial abundances are maximal when the water column is thermally stratified[16]. During stratification, eukaryotic phytoplankton are primarily seen at the deep chlorophyll maximum (DCM), which is in proximity to the nitricline[12,17]. Some of the general eukaryotic groups persisting in the DCM during the summer stratified periods have been hypothesized to be the same taxa that are observed in the winter mixing period at BATS, but data at even the genus level has so far been lacking[17], precluding further examination of this hypothesis.

For decades, the Sverdrup Hypothesis has provided a cornerstone for plankton ecology; it defined key interactions between biology and physics that shape late spring bloom dynamics[18]. However, in the subpolar Atlantic it is now recognized that spring bloom production initiates much earlier than previously thought, as taxa that are well-adapted to the lower light and potentially relaxed predation of the mixing regime populate the DM water column[9]. In one perspective, environment filtering explains early spring bloom dominance by a few taxa, but alternatively, as mentioned above, a study that observed prasinophyte algae in the bloom period also suggested that colonization of the DCM during summer might convey an advantage that contributes to their success in the early spring bloom[17]. Testing this hypothesis requires the taxonomic resolution to identify specific lineages that succeed both in the DCM and the early spring bloom – at a resolution beyond the phytoplankton species level.

Prasinophyte algae, particularly picoeukaryotic species (≤2 μm diameter), have been challenging to study due to morphological similarities and small cell size, as well as limited diagnostic or geologically preservable structures. While overlooked by many prior oceanographic methods, these unicellular green algae are ubiquitous in molecular surveys of the oceans[15,19–21], including cell enumeration by fluorescence in situ hybridization specific for prasinophytes[22]. They are now known to form a major fraction of the western North Atlantic spring bloom (north of 40°N), which had classically been attributed to diatoms[23]. At BATS, the Class II prasinophytes (i.e., Mamiellophyceae) *Micromonas* and *Ostreococcus* were first reported in clone libraries[24]; subsequently, qPCR from a handful of profiles indicated that these two picoplanktonic genera, and a third, *Bathycoccus*, had higher abundances during deep convective mixing periods[17]. In the subtropical North Pacific, at Station ALOHA, where convective mixing does not extend beyond the depth of the euphotic zone, these three species are predominantly seen at the DCM[25]. However, classifications, even at the level of genera and species, likely connect only loosely to ecological differences, as they do not encapsulate population biology or genomic variation that likely underlie population structure and niche differentiation[26,27]. To close this knowledge gap, ocean time-series studies that go beyond species-level characterization are critical; they provide the opportunity to pursue dynamics within environmentally contextualized baselines and annual cycles[4,28–30].

To date, little data exists that resolves prasinophyte dynamics across the euphotic zone on seasonal timescales. Indeed, the prasinophyte collective has not been studied in a temporal context in the open ocean. Here, we investigated prasinophyte dynamics over multiple years at BATS to characterize niche and seasonal transitions of different taxa and their overall contribution to phytoplankton communities of the open ocean. We examined prasinophytes at the species and amplicon sequence variant (ASV) levels using 16S rRNA gene amplicons, which have more constrained copy numbers that the 18S rRNA gene, improving representation of the relationship between relative amplicon abundance and cellular abundance[15,31]. Rather than working with set seasons (defined by calendar date), we identified varying periods of water column stability (hereafter

stability periods) that reflected seasonality within the water mass: the annual deep convective mixing event (DM), Spring Transition (ST), Stratified Summer (SS), and Autumn Transition (AT). We investigated whether there is a consistent prasinophyte signal associated with the DM and, if so, which taxa contribute to the bloom period, which prasinophytes are represented across the rest of the stability periods, and how they are structured vertically. The specific hypotheses tested were 1) prasinophytes are important contributors to the phytoplankton community during annual winter/spring bloom periods, 2) the same prasinophyte taxa found distributed throughout the mixed layer during the mixing period also inhabit the DCM during stratified periods[17], and 3) Class VII prasinophytes (i.e., Chloropicophyceae) are abundant open ocean prasinophytes, as implicated in a prior 18S rRNA amplicon-based study using normalization to photosynthetic eukaryote amplicons[20]. Finally, we sought to qualitatively consider the extent to which small eukaryotic phytoplankton contribute to export.

## Results

### Water column structure and chlorophyll concentrations

The BATS euphotic zone showed differences in water column stability affiliated with strong seasonality over our study period. Temperature variation in the euphotic zone demonstrated seasonal thermal stratification (Fig. 1b). Euphotic zone salinity patterns were less regular, with longer periods of lower salinity during the SS and AT but also shorter low-salinity periods during the DM (Fig. S1a). SS surface nutrient concentrations were often below detection limits (Fig. S1c-d, Supplementary Data S1, Supplementary Data S2). Surface temperatures ranged from 20.86 ± 0.75 °C (DM) to 21.09 ± 0.67 °C (ST) and reached a high of 27.33 ± 2.02 °C (SS) then decreased to 24.15 ± 1.40 °C (AT). The average temperature of the DM ML was 20.54 ± 0.72 °C. The highest surface chlorophyll *a* (Chl *a*) concentrations were recorded during the DM/ST, and the signal extended throughout the euphotic zone (Fig. 1c). The deepest mixing observed was in early April 2017, with the maximal mixed layer extending to 300 m, a depth that may be commonly reached but was captured herein due to cruise timing that intersected the event.

A systematic method was used to identify the DCM to facilitate interannual comparisons and distinguish analogous regions of the water column in the vertical during the DM period (when no DCM was present). We compared communities at ~1% PAR (during DM) to those in the identified DCM and to surface communities. During stratified conditions, the DCM ranged from ~80 m to 120 m, with an average temperature of 20.74 ± 1.03 °C. At the DCM, nitrate+nitrite concentrations were greater than the SS surface but highly variable, averaging 0.413 ± 0.466 μmol kg⁻¹ for nitrate+nitrite and 0.012 ± 0.031 μmol kg⁻¹ for phosphate (Fig. S1c-d). During DM, the 1% irradiance level was between ~80 m to 120 m and ML temperature averaged 20.54 ± 0.72 °C. Nutrient concentrations were similar to the DCM, 0.520 ± 1.631 μmol kg⁻¹ for nitrate+nitrite and 0.02 ± 0.074 μmol kg⁻¹ for phosphate.

At the surface, Chl *a* averaged 0.159 ± 0.092 μg kg⁻¹ (DM), 0.076 ± 0.060 μg kg⁻¹ (ST), 0.037 ± 0.015 μg kg⁻¹ (SS), and 0.068 ± 0.026 μg kg⁻¹ (AT). Surface Chl *a* was significantly higher at the DM than SS (Kruskal-Wallis and Dunn test statistic = −4.629, $p < 0.001$) (Fig. 1c). Significant differences in Chl *a* were not detected between the DCM (0.218 ± 0.136 μg kg⁻¹) and outside the stratified period at the ~1% irradiance level (0.172 ± 0.088 μg kg⁻¹; Kruskal-Wallis test statistic= 2.515, $p = 0.113$). The DM in vivo Chl *a* signal extended to ~250 to 300 m when we were on station during a DM event in 2017, while in the other years its detection was extinguished around ~160 m (Fig. 1c). Chl *a* and POC concentrations were positively correlated (Spearman ρ = 0.497, $p < 0.001$).

### Advanced methodology for prasinophyte species identification

The phylogenetic reconstruction developed herein utilized the 16S rRNA gene and covered the diversity of chlorophyte and prasinophyte algae, as well as prasinodermophytes (Fig. S2). Test data was placed correctly to the species level, however for two BATS *Micromonas* ASVs the 16S rRNA gene sequence had not been connected to 18S rRNA used to define species

**Fig. 1 | Sampling and oceanographic conditions at the BATS site from 2016 to 2020. a** Sampling locations in the BATS site's proximity superimposed over blended 5 km resolution night sea surface temperature data (shown for December 2019; National Oceanic and Atmospheric Administration CoastWatch). The number of CTD profiles for DNA samples collected at a location is indicated (bubble size), with the inset showing a zoom of the BATS region. **b** Temperature (°C), (**c**) Chl *a* concentration (µg kg⁻¹) from 12 depths from the surface to 250 m, and (**d**) percentage of amplicons assigned to the prasinophytes out of all plastid-derived amplicons (16S V1-V2) over the course of the study, based on interpolation from discrete data points (black dots; 8 depths per profile for **c** and **d**, and interpolation on the horizontal between continuous measurements over depth for **b**). Superimposed over (**b-d**) are lines indicating the DCM in black and mixed layer depth (MLD) in white. These are defined in **b**, as are the water column stability periods (SS stratified summer, AT autumn transition, DM deep mixing, and ST spring transition) indicated by the green bar in **b–d**.

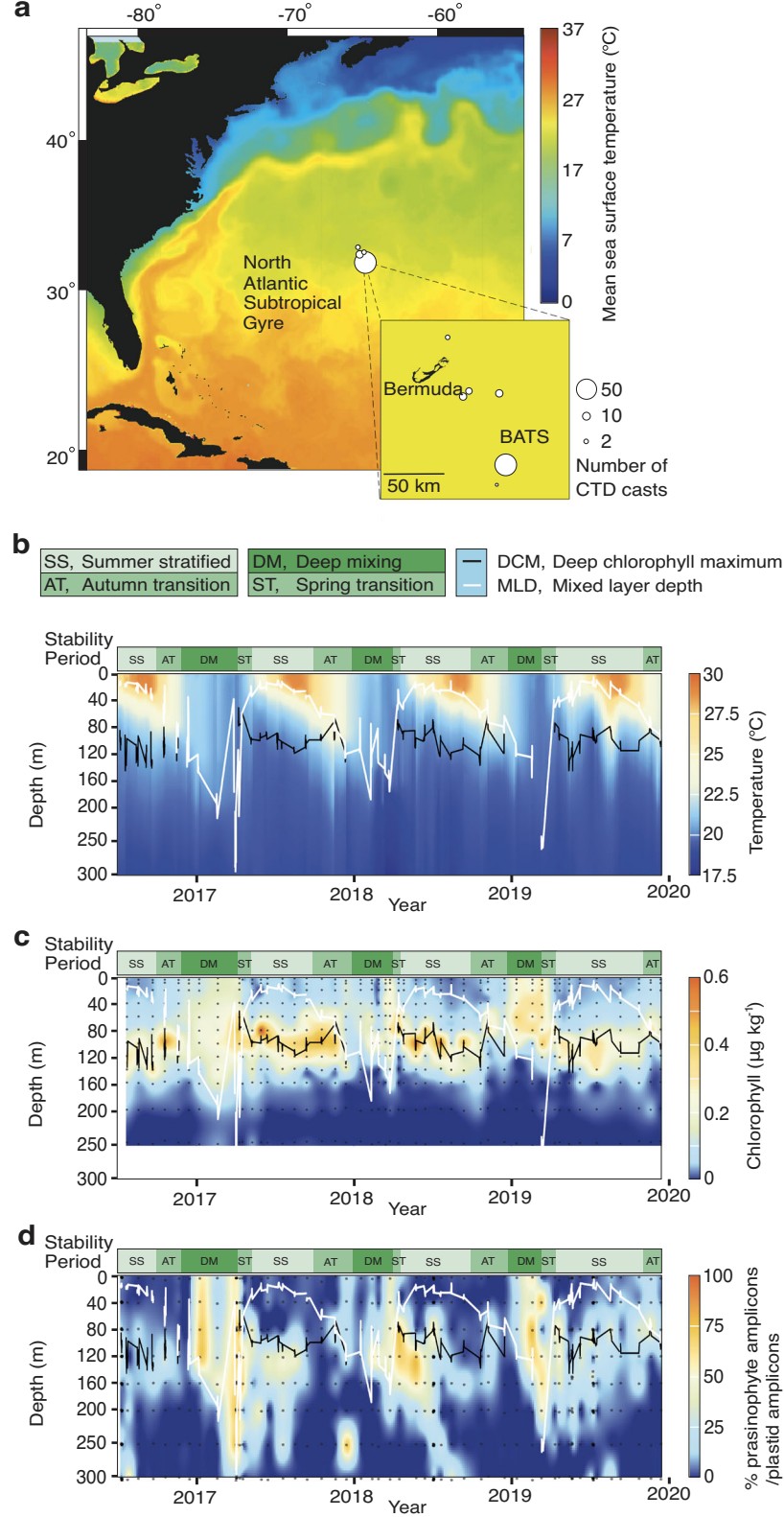

because the species (plural) are uncultured. Therefore, we compared proportions of these ASVs to 18S data[32] available for a subset of the samples using statistical approaches (Fig. S3c). For ground truthing of this approach, *Micromonas commoda sensu stricto* (Clade A as in[27,33], which includes the type species strain RCC299; hereafter *M. commoda ss*) amplicons from 16S rRNA and 18S rRNA data were compared first and found to be correlated

(Spearman test, ρ = 0.650, *p* < 0.001). The same approach rendered a correlation (Spearman test, ρ = 0.856, *p* < 0.001) between ASV81 and 18S sequences from uncultivated Clade B._.4, which was first reported in the North Pacific[34] (now termed *Micromonas* candidate species 1[26]). Similarly, *Micromonas* 16S ASV1156 was significantly correlated with *M.* candidate species 2 18S rRNA ASVs (Spearman test, ρ = 0.802, *p* = 0.001). These

results indicated ASV81 and ASV1156 could represent *Micromonas* candidate species 1 and candidate species 2, respectively.

Of sequences with LCA-best node mismatches, 15 had 90-99% identity to environmental clones from the North Pacific, six with >98% nucleotide (nt) identity to these clones and a second related group of nine ASVs (Fig. S4). The complete 16S rRNA gene clone sequences from the North Pacific were then used as queries themselves, recovering DQ438491 (98% nt identity) from the East China Sea. DQ438491 is linked in PR$^2$ to uncultivated prasinophyte Class IX 18S rRNA sequences. Although we therefore tentatively termed the 15 ASVs putative Class IX, we did not recover a significant relationship between the 16S ASVs and Class IX 18S rRNA ASVs (as per PR$^2$) from the 140 BATS samples in which they were compared. After these refinements, our phylogenetic approach indicated all prasinophyte classes (including the Prasinodermophyta) except Class III are present at BATS.

### Prasinophyte euphotic zone contributions over stability periods and depths

Analysis of the molecular diversity of algal communities using plastid-derived V1-V2 16S amplicons from 79 profiles (8 depths from the surface to 300 m) revealed trends of prasinophytes at BATS over the annual cycle. Amplicon analyses indicated that periods of prasinophyte highest relative abundances (among eukaryotic phytoplankton) corresponded with overall patterns in Chl *a* (Fig. 1c, d). During DM prasinophytes averaged 46.3 ± 24.2% of plastid amplicons in the mixed layer (Fig. S5), while prymnesiophytes averaged 18.3 ± 8.2% of plastid amplicons, stramenopiles made up 31.5 ± 21.2%, and the remaining 3.1 ± 2.2% was from other eukaryotic phytoplankton groups. At the surface (~1–5 m), prasinophyte contributions to plastid amplicons ranged from 33.6 ± 24.6% (DM), to 8.8 ± 14.5% (ST), to 3.6 ± 3.4% (SS), and 6.6 ± 7.9% (AT), with significant differences between the DM and the SS (Kruskal-Wallis and Dunn test statistic = −4.046, $p < 0.001$) (Supplementary S2). Prasinophytes at the DCM averaged 29.0 ± 21.7% out of plastid amplicons, with prymnesiophytes making up 19.3 ± 7.9%, stramenopiles 50.1 ± 17.4%, and the remaining 1.5 ± 1.4% from other groups. The percentage of prasinophytes out of plastid amplicons was positively correlated ($p < 0.001$) to Chl fluorescence (Spearman $\rho = 0.397$) and Chl *a* (Spearman $\rho = 0.252$) (Supplementary Data S3). Overall, the entire DM mixed layer and the DCM had the highest relative prasinophyte contributions out of the identified stability periods and water column zones.

We compared prasinophyte communities in the surface (~1–5 m) to those of the DCM and ~1% irradiance (during DM). This revealed annual cycles in the relative abundance of different genera and classes in the surface waters as well as at depth (Fig. 2). During the DM and ST, when surface Chl *a* concentrations were highest, Class II, specifically *Bathycoccus*, *Micromonas*, and *Ostreococcus*, had the greatest prasinophyte contributions, particularly the latter two. Apart from *Micromonas* during 2019, these genera were low or undetected in amplicon data from the surface during SS (Fig. 2b). Chl *a* concentrations during SS were extremely low in surface waters, and the prasinophytes with the highest relative abundances belonged to the Class I Pyramimonadales and unclassified prasinophytes, putatively identified as Class IX (Fig. 2a).

The distributions of prasinophyte genera in the DCM, or at the ~1% irradiance level during mixing periods, contrasted with those seen in the surface 5 m, particularly the SS surface. Class II *Bathycoccus* and *Ostreococcus* were detected throughout the year at these lower depths, with *Ostreococcus* generally having the highest relative abundances (Fig. 2b). *Micromonas* had elevated relative contributions during DM periods at the 1% light level and was frequently detected at the DCM. Class VI (Prasinodermophyta), Class I Pyramimonadales, Class VII *Chloropicon* and *Chloroparvula*, and prasinophytes putatively belonging to Class IX were detected sporadically at the DCM (Fig. 2b).

### Greatest Class II prasinophyte contributions correspond with high chlorophyll concentrations

Having observed patterns of seemingly coincident higher Chl *a* and prasinophytes in the mixed layer during DM and in the DCM during stratified periods (Figs. 1, 2), we next examined these patterns at greater depth resolution. To this end, we expanded our analyses from the surface and DCM, or 1% light level, samples alone, to all samples from the upper 140 m (the base of the euphotic zone). Quadrants were defined by 75$^{th}$ percentile quartiles for each of these measurements (Fig. 3). Chl *a* was significantly different between quadrants except Q2 (>75$^{th}$ percentile Chl *a*, <75$^{th}$ percentile prasinophyte) and Q4 (>75$^{th}$ percentile Chl *a*, >75$^{th}$ percentile prasinophyte). The percent prasinophytes between each quadrant was significantly different except between Q3 (<75$^{th}$ percentile Chl *a*, >75$^{th}$ percentile prasinophyte) and Q4 (>75$^{th}$ percentile Chl *a*, >75$^{th}$ percentile prasinophyte). The Q1 quadrant (<75$^{th}$ percentile Chl *a*, <75$^{th}$ percentile prasinophyte) differed the most compositionally, wherein Class II members still comprised the majority, but with notable contributions from Class I and putative Class IX prasinophytes. In contrast, Q2, Q3, and Q4 mostly comprised specific Class II species, i.e., *Bathycoccus calidus*, *Ostreococcus* Clade OII, *M. commoda ss*, and *Micromonas* candidate species 1 (Fig. 3b).

### Niche partitioning of Class II prasinophyte sub-species variants

*Bathycoccus*, *Micromonas*, and *Ostreococcus* accounted for up to 85.6% and 46.1 ± 24.6% on average of plastid-derived amplicons in the mixed layer during DM (Figs. 1, 2, Supplementary Data S1). To examine patterns across the vertical dimension, we normalized prasinophyte amplicons by the total number of 16S amplicons (i.e., from all heterotrophic bacteria, cyanobacteria, and plastids) to diminish misleading signals from samples with very few phytoplankton. During DM *Bathycoccus*, *Micromonas*, and *Ostreococcus* formed up to 17.9% and on average 2.9 ± 3.1% of total 16S amplicons. Fleeting appearances of Clade OI *Ostreococcus lucimarinus* (two ASVs averaging 0.09 ± 0.07% of total amplicons in 7 samples out of 431) and *Bathycoccus prasinos* were observed (0.02 ± 0.02% of total amplicons in 4 samples; Supplementary Data S1). The majority of *Bathycoccus*, *Micromonas*, and *Ostreococcus* relative abundances were formed by a few sub-species variants. For example, together, *Ostreococcus* OII ASV6 and ASV77 accounted for 96.4% of *Ostreococcus* amplicons across all samples, and *B. calidus* ASV177 accounted for 98.1% of all *Bathycoccus* amplicons.

The dominant sub-species variants exhibited patterns associated with water column stability periods and vertical stratification zones. *Ostreococcus* Clade OII ASV6 was well-represented in the DM and DCM (Fig. 4a), and ASV77 had a similar distribution but with generally lower relative abundances than ASV6, except for in the AT (Fig. 4b). *Bathycoccus calidus* ASV177 had broadly similar patterns to *Ostreococcus* Clade OII dominants, transitioning from the DM ML into the DCM once the system stratified (Fig. 4c). While ASV6 had a clear surface signal in all three DM periods, ASV77 and ASV177 were well-represented throughout the euphotic zone during the DM period of 2017 but had low relative abundances in 2018 and 2019 DM periods.

The dominant *Micromonas* species at BATS was *M. commoda ss* (Figs. 3, 4d-e). *Micromonas* candidate species 1 also exhibited high relative abundances (Figs. 3, 4f). Two ASVs belonging to *Micromonas bravo* (Clade E1) were detected, ASV503 and ASV12724. The former was found only in 2017 in the DM, ST, and SS between 80-200 m, while the latter was found from 10-80 m across all years in either summer or autumn (Supplementary Data S1). Two *Micromonas pusilla* (Clade D) ASVs were detected, at 250 m during the 2017 DM and 120 m during the 2019 SS (Supplementary Data S1). ASV61, a sub-species variant of *M. commoda ss*, and *M.* candidate species 1 ASV81 together contributed 80.2% of all *Micromonas* amplicons. While similar in reaching highest relative abundances during DM, ASV81 exhibited diminished importance during the 2018 DM (Fig. 4f). Comparison of other *M. commoda ss* sub-variants to ASV61 highlighted interannual differences in sub-variant patterns. For instance, ASV273 and ASV345 had a strong presence in the 2017 DM, but not the DM periods of 2018 and 2019 (Fig. 4d-e). Moreover, *Micromonas* candidate species 2 ASV1156 displayed a different seasonal pattern from the more abundant Class II ASVs, reaching its maximum relative abundance within the upper 120 m ML during the

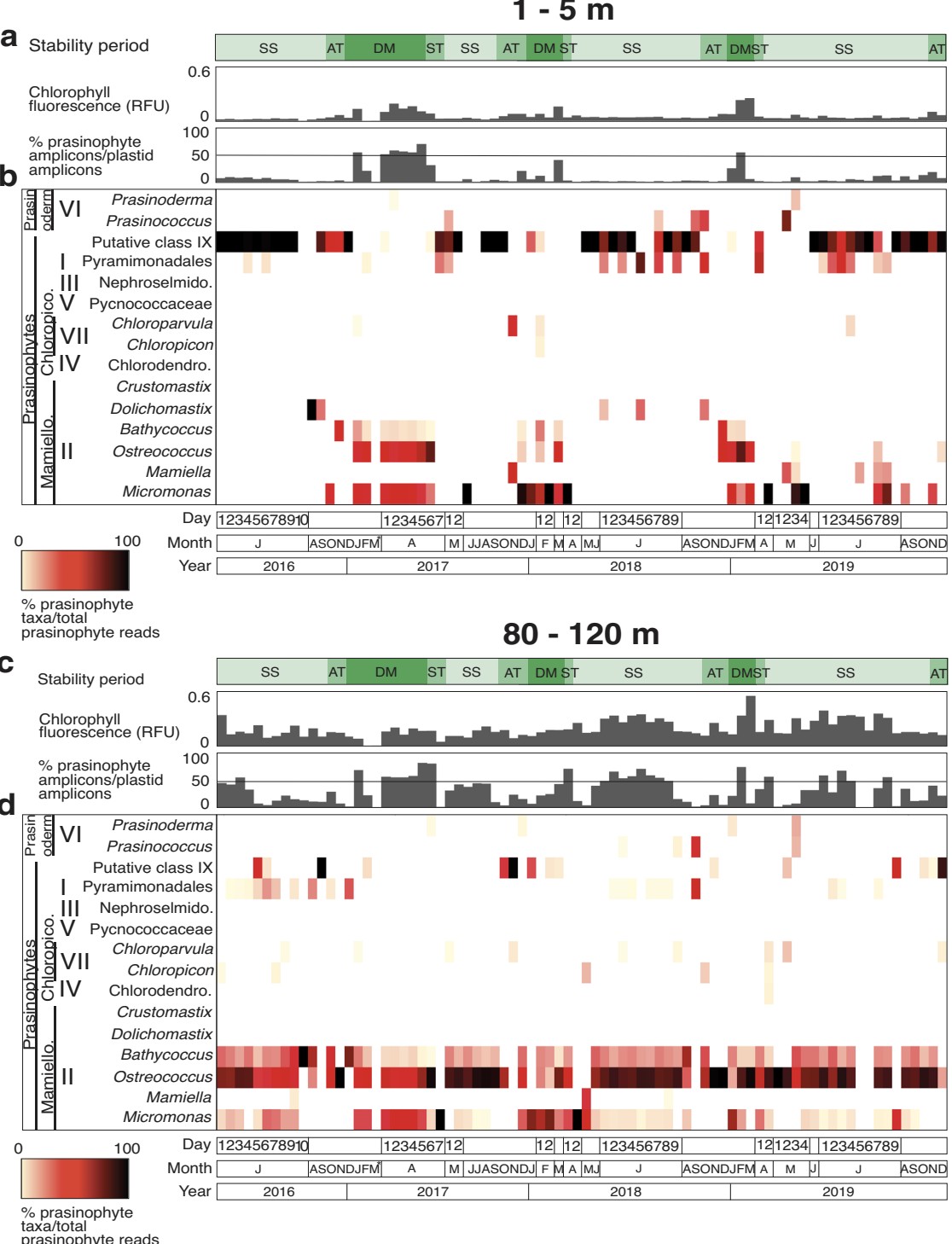

**Fig. 2 | Prasinophyte distributions in the surface 5 m and between 80 – 120 m across time and varying states of thermal stability.** Data is shown from July 2016 to December 2019. Above each major panel the top bar plot shows in vivo Chl *a* fluorescence (RFU) from the exact sample analysed (chlorophyll concentrations were also measured, but sometimes from a different cast the same day). The plot below shows percentage of prasinophyte amplicons out of plastid amplicons with a line indicating the 50% contribution level. These parameters are plotted for (**a**) the surface 5 m and (**c**) between 80 and 120 m, the proximity of the DCM during stratified periods. The heatmaps show relative abundance of different prasinophyte taxa in relation to the number of prasinophyte sequences in the sample for (**b**) the surface and (**d**) the DCM and ~1% irradiance during mixing periods. Note the X-axis represents sampling dates and is not scaled linearly according to time, due to heavier sampling during highly dynamic periods. Sampling did not take place in March 2017 (asterisk). Water column stability periods (SS stratified summer, AT autumn transition, DM deep mixing, and ST spring transition) are indicated by the green bar.

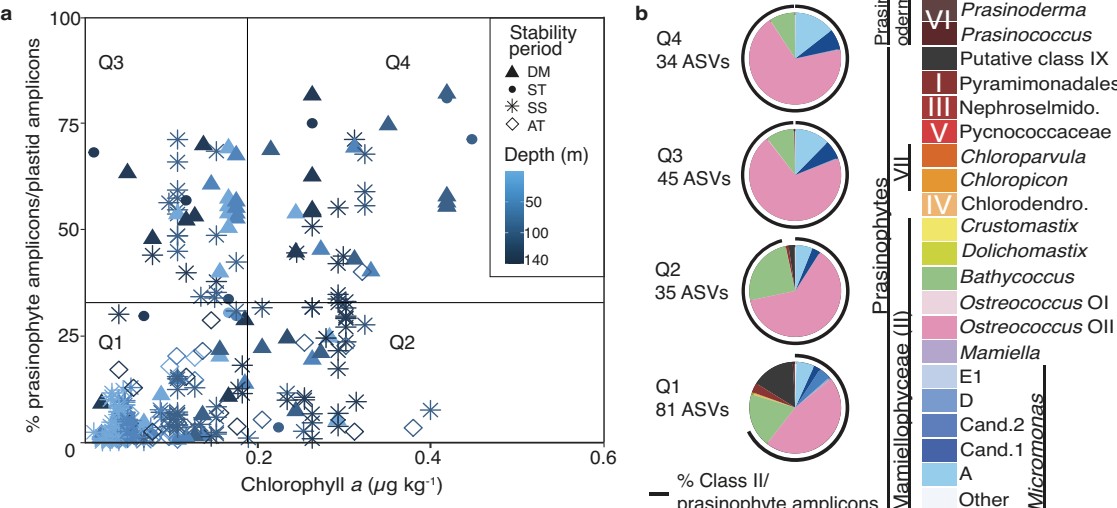

**Fig. 3 | Relationship between Chl *a* and green algal contributions to the eukaryotic phytoplankton community. a** Chlorophyll *a* (μg kg⁻¹) versus percent prasinophytes out of plastid amplicons. Symbol shape corresponds to stability period and color corresponds to depth. The plot is divided into quadrants by Chl *a* values (above and below 0.188 μg kg⁻¹, the 75th percentile Chl *a* value at BATS in the surface 140 m from June 2016 to December 2019) and by the 75th percentile of prasinophyte amplicons out of plastid amplicons (33.2%). **b** Prasinophyte group contributions to prasinophyte amplicon counts per quadrant. The number of ASVs per quadrant is indicated, and the black semi-circle around the pie charts indicate the contribution of Class II to prasinophyte ASVs.

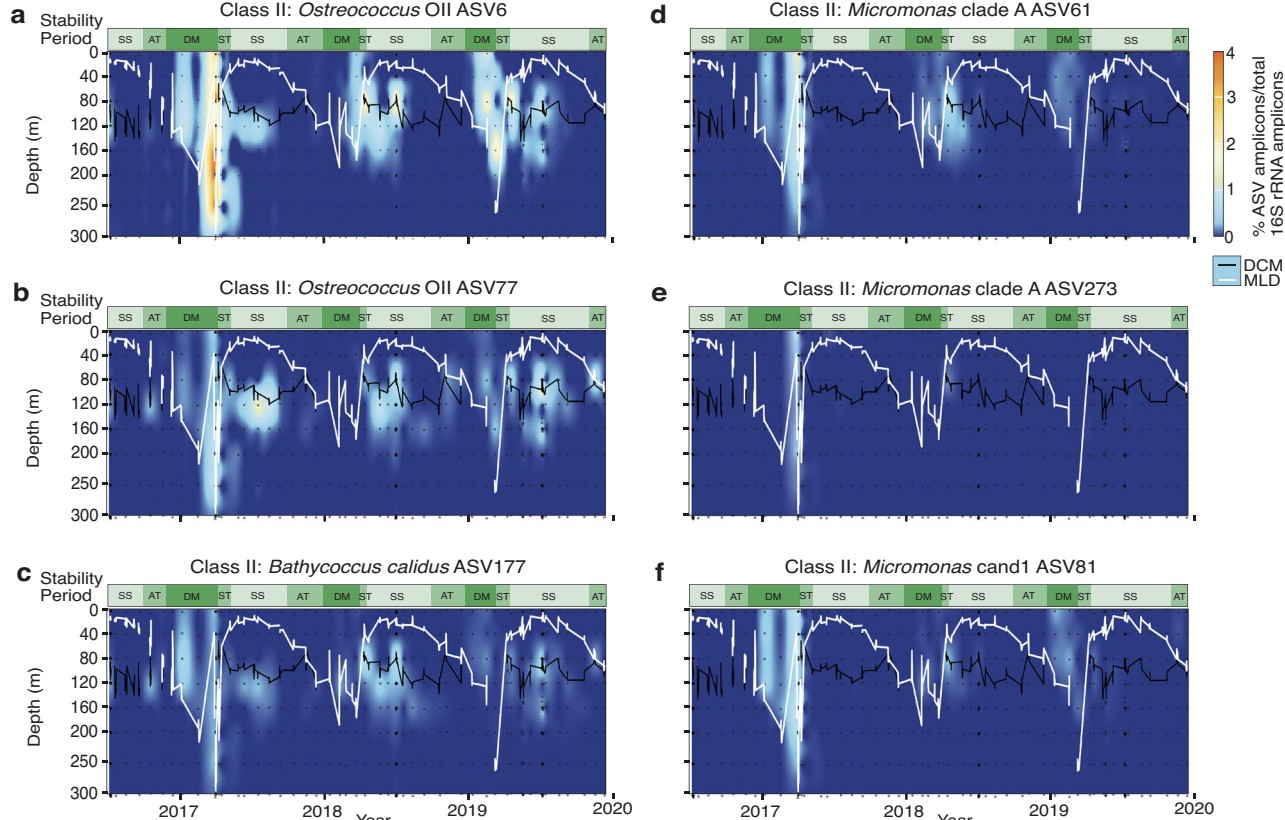

**Fig. 4 | Distribution of highly abundant prasinophyte Class II ASVs.** Relative abundances of (**a**, **b**) *Ostreococcus* OII (ASV6, ASV77), (**c**) *Bathycoccus calidus* (ASV177), (**d**, **e**) *Micromonas commoda* ss (ASV61, ASV273), and (**f**) *Micromonas* candidate species 1 (ASV81) out of total amplicons (including all bacteria and plastids) in the upper 300 m of the water column from July 2016 to December 2019. Data presentation is interpolated from discrete data points (black dots; 8 depths per profile). Water column stability periods are as in prior figures. Superimposed are lines indicating the DCM (black) and MLD (white).

**Fig. 5 | Relationship of prasinophyte ASVs to environmental variables.** Partial canonical correspondence analysis plot of prasinophyte ASVs with water column stability period as the conditioning variable, with vectors representing temperature (°C), salinity, nitrate + nitrite (µmol kg⁻¹), and phosphate (µmol kg⁻¹). The proportion of variation (inertia) explained by the constrained eigenvalues is indicated in parentheses for each axis. Vectors with an asterisk indicate that the variable was significantly correlated with the variation in the prasinophyte ASVs. Prasinophyte groups are indicated by color, with several important ASVs labelled by number.

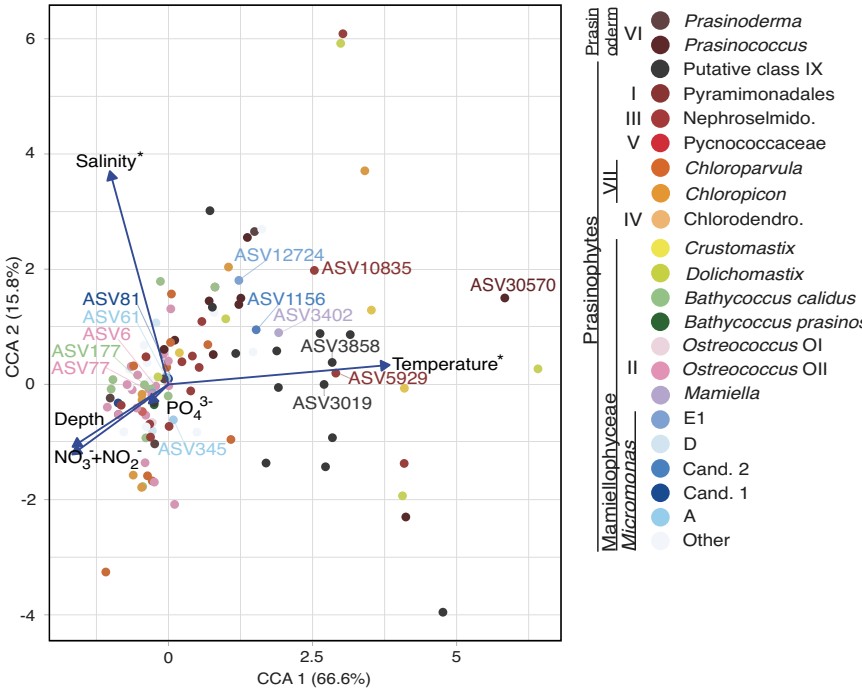

## Persistent and ephemeral sub-species variants are differentially distributed over stability periods

The overall ASV-level composition of the prasinophyte community across years, stability periods, and depths was significantly associated with temperature and salinity variability (permutation test for canonical correspondence analysis (CCA) using abiotic factors, p < 0.001; Fig. 5). A partial CCA (pCCA) incorporating Chl *a* data (which at least in part would be derived from the prasinophyte algae) showed that Class II grouped closer to the Chl *a* vector than many other prasinophyte groups, however, unlike temperature and salinity, the association was not significant (Fig. S7). Some prasinophyte ASVs were associated with higher temperatures, such as Class II *Micromonas* candidate species 2 ASV1156, *Mamiella* ASV3402, Class I ASV5929 and ASV10835, and putative Class IX ASV3019, ASV3858, and ASV2562. ASV-level prasinophyte community composition did not vary significantly with depth within the ML during DM (ANOSIM test statistic r = −0.08, *p* = 1); however, the DM ML prasinophyte community was statistically different from that of the SS surface ML (ANOSIM test statistic r = 0.76, *p* < 0.001) and SS DCM (ANOSIM test statistic r = 0.21, *p* < 0.001; Supplementary Data S3).

Some prasinophytes demonstrated strong partitioning connected to stability periods and depth, while others were persistently present (Fig. 6a,b, Supplementary Data S1). Persistent sub-species variants exhibited greater overlap between years (Fig. 6c) and included some taxa with low relative abundances, others with high relative abundances, and with seasonal, vertical, or interannual variations (see Supplementary Note 1). Several persistent Class II sub-species variants had the highest relative abundances among prasinophytes—specifically, *Ostreococcus* Clade OII ASV77 and *B. calidus* ASV177 (Fig. 6, Fig. 4). These ASVs were abundant throughout the euphotic zone during DM and ST and then present at the DCM during SS and AT.

The stability period with the highest number of ASVs in the euphotic zone (74 in total from 140 m and above) was the SS (Fig. 6a,b). Forty-three of these were exclusive to the SS, which was the largest proportion of ephemeral ASVs detected only in one stability period (see Supplementary Text). Several Classes contributed during summer, and some fairly consistently like Class I and Class VI *Prasinococcus* ASVs, which together

2017 AT. It was also detected at various depths during the DM periods of 2017, 2018, and 2019 and in the SS surface in 2019 (Figs. S6b and 6a).

comprised 0.57 ± 2.4% of SS plastid amplicons (Fig. S6, Supplementary Data S1). Ephemeral SS sub-species variants also exhibited depth variations. Three SS 'all years' ASVs (*Ostreococcus* OII ASV2949 and ASV33032, *M.* candidate species 1 ASV81) were found only at or below the DCM, alongside other depth patterns for other taxa, like *M.* candidate species 2 ASV1156 (Fig. S5, S6b). Finally, several non-dominants contrasted with distributional patterns of the dominant ASVs from that species, while for other groups, seasonal trends were relatively consistent. Thus, exclusively SS ephemerals, as well as some other ephemerals exhibited depth, seasonal, and interannual variation (Supplementary Text).

## Discussion

Picophytoplankton are extremely important to global primary production due to their dominance in tropical and subtropical ecosystems[8]. Although cyanobacteria, especially *Prochlorococcus,* are highly abundant in these ecosystems[12,16,35], picoeukaryotes were shown to be important to primary production in the subtropical North Atlantic over two decades ago [69]. Prior amplicon-based time series studies at BATS[32], the North Pacific subtropical gyre at HOT[28], and more coastal Pacific San Pedro Ocean Time-Series (SPOT)[29] have noted prasinophytes, particularly Class II[36,37]. However, although seasonal transitions in open ocean environments are known to cause changes in MLD, concentrations of limiting nutrients, and overall vertical structure of the euphotic zone[5,12], how picoeukaryotic species and populations respond has remained unknown. This knowledge gap arises from three primary issues: i.) The small size of picoeukaryotes makes morphological identification of individual genera or species virtually impossible via microscopy, such that only the advent of molecular approaches has allowed consistent identification. ii.) Often amplicon-based surveys of phytoplankton communities largely comprise surface water samples (top 10 m) or lack vertically-contextualized sampling, missing depths where these cells might thrive. iii.) Weather related challenges to sampling have led to biases for more quiescent times of the year. Therefore here, we performed systematic time- and depth-resolved sampling to determine successional patterns of these algae over defined stability periods in the North Atlantic subtropical gyre at the BATS site. Our study was designed with the growing recognition that understanding ecological dynamics and evolutionary trajectories requires resolution of species and sub-species variants. Resolution at this level reflects functional diversity

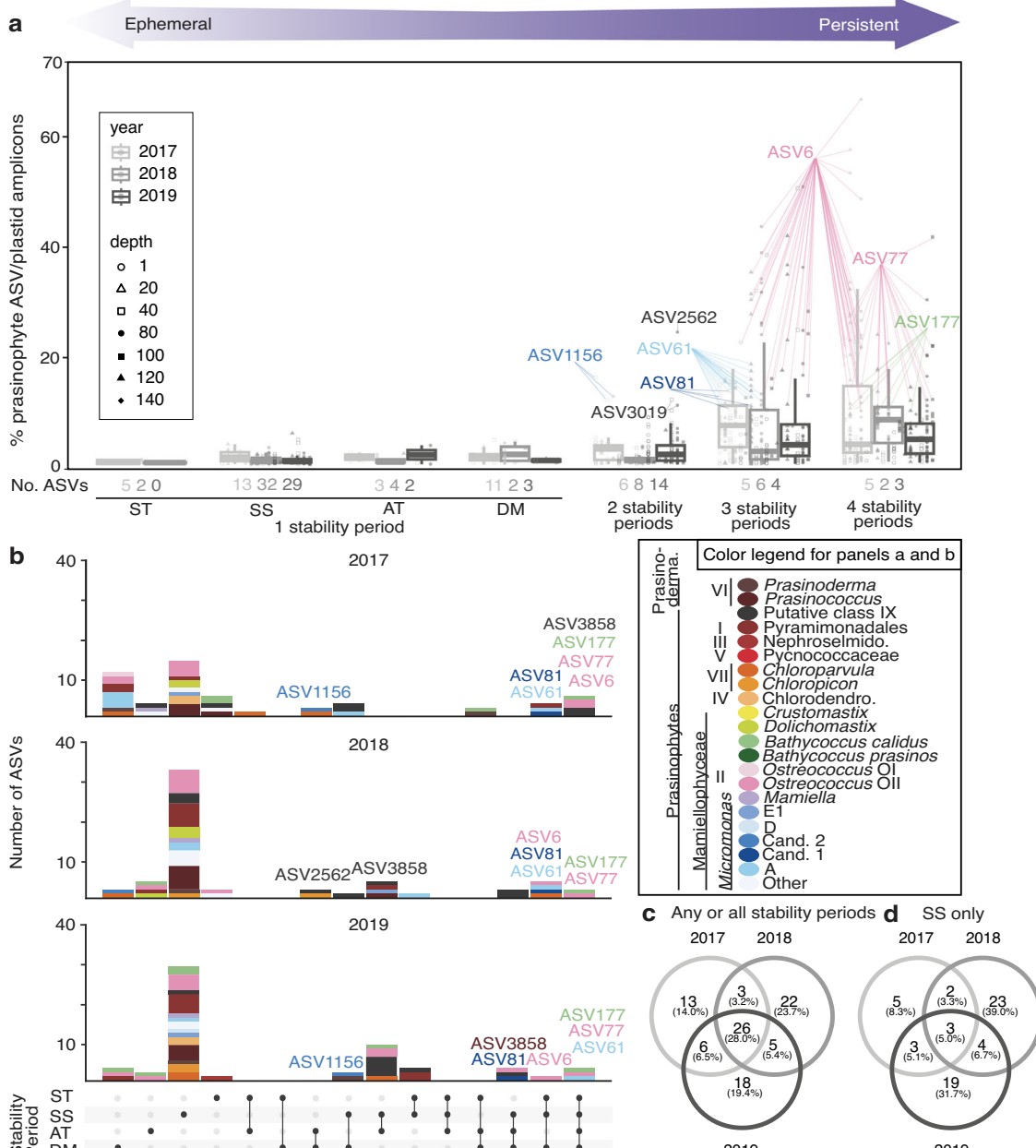

**Fig. 6 | Ephemeral and persistent green algal sub-species variants or ecotypes.**
**a** Relative abundance of prasinophyte sub-species variants out of plastid amplicons in samples (symbols, also indicating depth) and stability periods (i.e., detected in only one or in multiple) across years (light to dark grey) where these ASVs were found. Sub-species variants forming ≥10% of plastid amplicons in individual samples are labelled with the ASV number with colors as in panel b. The box represents the median and interquartile range and whiskers the greatest and least non-outlier points. Singletons were excluded. **b** UpSet plots show ASV distributions by class or species (colors) and stability period (a black dot on the row indicating stability period means ASVs in the bar graph above detected in that period). **c** Venn diagram of prasinophyte ASVs in any or all stability periods showing overlap between years. **d** Venn diagram of prasinophyte ASVs as in panel c, but found only in the SS stability period. Note that biases in sampling frequency within a stability period can influence the observed patterns, e.g., fewer profiles were collected during AT than other periods. Note that for all panels only data from 140 m and above is used and 2016 is excluded due to incomplete sampling of the annual cycle.

more so than broader microbial classifications[4,38,39]. Here, we show major contributions of prasinophytes in the open ocean during more dynamic periods alongside marked shifts in community structure and key players over the annual cycle, contributions and patterns that have been missed in the absence of vertical- and time-resolved sampling.

## Annual cycles and the emerging importance of prasinophytes in the open ocean

NPP at BATS is maximal within the surface 80 m of the DM and into the ST periods compared to other parts of the annual cycle[6]. However, it should be

noted that between 2010 and 2020 decreased estimates of 0-140 m integrated primary production were observed[40], potentially indicating ecosystem-wide changes to the phytoplankton community. We evaluated the community at BATS between July 2016 and December 2019 (Fig. 1 and S1) over euphotic zone profiles in which annually recurring biogeochemical and chlorophyll variations were similar to previous reports at this site[6,17] and established the presence of strong seasonal transitions at BATS. The demonstrated patterns associated with a vigorous deep mixing period in the winter/early spring were followed by a stable stratified period in summer/early autumn.

Ordination analysis indicated prasinophyte ASV distributions were influenced by density discontinuities, which connected to the stability periods (Fig. 5) and Chl *a* distributions (Fig. 1c). Chl *a* concentrations were highest during DM throughout the ML, and at the DCM for the rest of the annual cycle when the ML shoaled to depths shallower than the 1% light level (Fig. 1c and S1c). Chl *a* was positively correlated to prasinophyte contributions to eukaryotic phytoplankton (Fig. 1d). While differences in size and distribution of phytoplankton taxa underpin phytoplankton biomass and influence the relationship between Chl *a* and Chl fluorescence per cell, the relationship between Chl *a* and biomass is not straightforward especially across vertical profiles[41]. We note also that some taxa, such as dinoflagellates, have unique plastids[42] that may not be effectively represented by the 16S rRNA PCR approach employed here, and that for all organisms the state of genome replication and proximity to division at the time of collection will influence numbers[31]. Here, Chl *a* and POC were correlated, as observed in some prior BATS studies[13,40,43]. Thus, at least during DM when strong photoacclimation is unlikely, Chl *a* presumably reflected distributions of phytoplankton biomass. This is supported by flow-cytometry-derived estimates of eukaryotic phytoplankton biomass, wherein eukaryotic phytoplankton reach their highest contribution to total phytoplankton biomass during the winter/spring[12,44]. The overall contributions of prasinophytes to total plastid amplicons also increased with increasing Chl *a* (Fig. 3), highlighting the likely importance of prasinophytes to PP.

There was a significant shift in prasinophyte community composition at the surface from the DM period into the SS (Figs. 2b, 4, S6). The DM dominants, all Class II, followed the trajectory of the nitricline as nutrients became unmeasurable in the surface 40–80 m during late spring stratification. The SS prasinophyte community exhibited Class I and putative Class IX prasinophytes extending to ~120 m, however, primary production by these classes was presumably low, due to low relative abundances among eukaryotic phytoplankton, and the dominance of cyanobacteria, particularly *Prochlorococcus*[12,16,44], under these conditions. Class II again was prominent in the SS DCM. These findings underscored still unknown aspects of niche differentiation that underpin shifts in prasinophyte classes seen across stability periods and depth zones at BATS.

### Resolving species and rectifying misidentification of prasinophytes

A hindrance to understanding prasinophytes is that they are polyphyletic, comprising nine divergent lineages (based on 18S rRNA gene phylogenies), one of which (Class VI) has recently been reclassified as its own phylum, the Prasinodermophyta[45]. Another Clade formerly identified in 16S rRNA gene data (Class VIII) has been proposed to be a subgroup within Clade VII (as identified in 18S rRNA gene data), specifically VII.B (*Chloroparvula*)[46]. Several species, including *B. prasinos* and *B. calidus*[47], cannot be delineated using commonly analysed 18S rRNA gene marker regions[45,48] and our results (see below) indicate reported distributions of '*B. prasinos*' based on 18S rRNA gene sequencing should be reevaluated. Putative Clade IX could not be definitively connected to an 18S rRNA-defined lineage perhaps in part due to its low abundance at BATS (Fig. S4). Further analyses will be needed to establish the evolutionary relationship between these uncultivated algae—which are present in the North Pacific subtropical gyre[49] and East China Sea—and other green algae. These results point to the importance of information from genomic and metagenomic analyses alongside full-length gene phylogenetic analyses and use of appropriate marker regions for gaining a more nuanced view of speciation and ecological distributions.

Finally, few studies have clearly partitioned the seven evolutionarily distinct lineages of *Micromonas* in environmental data. Herein, we consider all *Micromonas* to be photoautotrophs, as previously suggested predatory mixotrophy by *Micromonas polaris*[50] has been largely debunked[51,52]. Of the diverse species (Clades) of *Micromonas*, two remain uncultured and hence lacked information connecting 16S

and 18S rRNA gene data. Our comparisons of 16S and 18S ASVs from the same samples, alongside results from the Caribbean[53], identified 16S ASVs likely belonging to *Micromonas* candidate species 1 and 2 (Fig. S3).

### Stability period- and vertically resolved sampling highlight the importance of Class II prasinophytes in the open ocean

Our results provide insights on picoeukaryotes of the open ocean. Most of the prasinophyte ASVs that reached notable relative abundances came from Class II species (Fig. 3). Class II prasinophytes have often been considered coastal[20,54], but more than a decade ago the possibility of marked contributions to oceanic phytoplankton communities was reported in the DCM[55]. The temporally resolved data herein shows that particular Class II species dominate plastid relative abundances in many of the higher chlorophyll samples (Fig. 2, Fig. 3). The largest share of the prasinophyte contributions to all eukaryotic phytoplankton during DM is attributed to the Class II prasinophytes *Ostreococcus* Clade OII, *M. commoda ss*, and *M.* candidate species 1 (Figs. 2–4). *M.* candidate species 2, *M. bravo* (Clade E1), and *M. pusilla* (Clade D) are also present but with minor roles. After stratification, these taxa, as well as newly recognized *B. calidus*[47] remained important at the DCM (Figs. 2 and 3), with *B. calidus* being present across all years and stability periods.

Other studies have analysed datasets with a predominance of surface samples or non-vertically resolved samples. For example, an Ocean Sampling Day-based study including 18S V4 amplicon data from the surface stratified period at the BATS site identified Class I, Class VII, and Class II as the major prasinophyte groups[54]. Often such studies have reported percentages of these Classes out of total green algal amplicons, rather than the entire eukaryotic phytoplankton community, a practice which may overestimate their importance – especially if collected during periods where other taxa prevail, like *Prochlorococcus* in the stratified surface at BATS. Class VII has been observed in the South Pacific Subtropical Gyre[56] and a more general survey (*Tara* Oceans) suggested Class VII are the most important prasinophytes in the open ocean, contributing on average 8% of total photosynthetic eukaryote amplicons[20]. Differences in approaches to sequence analysis, specifically resolution in 18S rRNA V9 data[48] and *Tara*'s use of Swarms[57] make direct comparisons difficult. In our study, while both Class I and VII are present at BATS, especially in the upper 40 m during SS, they were detected intermittently and at low relative amplicon contributions (< 0.6% of plastid amplicons, annual average) to the eukaryotic phytoplankton community (Fig. S5b, S6a,f-g). The sporadic and low detection of Class VII contrasts with the third hypothesis, although comparative seasonally- and vertically-resolved sampling of other subtropical regions would be needed to further test the hypothesis. Overall, we find that although pronounced Class level distinctions have been described between coastal and open ocean prasinophyte communities, they largely come from studies that lack seasonally- and vertically- resolved sampling protocols in the open ocean.

Collectively, our findings on the importance of Class II align with more general data from flow cytometric enumeration showing picoeukaryote abundance is significantly correlated with periods of higher Chl *a* around the DM at BATS[58]. Prasinophytes were reported as early contributors to the winter/spring bloom at the BATS site between 1991 and 2004 based on T-RFLP and Class II 18S rRNA qPCR data, while other phytoplankton (cryptophytes, haptophytes, and pelagophytes) had their individual maxima later in the bloom period[17]. Moreover, in the subtropical North Atlantic northwards of BATS, amplicon analyses showed high relative abundances of Class II[23]. Finally, 18S rRNA qPCR-based studies demonstrate that both *Ostreococcus* Clade OII and *B. calidus* are present throughout the ML until waters stratify, and are then abundant at the DCM of BATS[17,55] and HOT station ALOHA[25]. Thus, in contrast to some prior ocean surface survey studies, our results, supported by more limited prior quantitative data, establish the importance of picoeukaryotic Class II prasinophytes in open-ocean environments.

**Table 1 | Global distribution of the dominant persistent sub-species variants at BATS**

| Class II Prasinophyte | | BATS | Bay of Bengal | Curaçao | N. Atl. Spring (5–100 m) | |
|---|---|---|---|---|---|---|
| Species | Sub-Species | <140 m | 2-100 m | 0-23 m | Subpolar | Subtropical |
| *Ostreococcus* Clade OII | ASV6 | 67 | 53 | 28 | n.d. | 3 |
| *Ostreococcus* Clade OII | ASV77 | 42 | 78 | 8 | n.d. | * |
| *Micromonas commoda ss* | ASV61 | 18 | 5 | 24 | n.d. | n.d. |
| *Bathycoccus calidus* | ASV177 | 17 | 10 | 10 | n.d. | * |
| *Micromonas* cand. sp. 2 | ASV1156 | 16 | * | 12 | n.d. | n.d. |
| *Micromonas* cand. sp. 1 | ASV81 | 14 | n.d. | 22 | n.d. | 4 |

Only sub-species variants that exceeded 10% of total plastid amplicons in at least one euphotic zone sample (0 to 140 m) at BATS are shown (with value represented as "maximum percentage of plastids"). Non-BATS data comes from [23,53,59]; note that table only includes ASVs from these papers detected here at BATS. Curaçao indicates Southern Caribbean Sea sampled near Curaçao, n.d. indicates not detected, and * indicates <0.5% plastid amplicon abundance.

## Dominant picoprasinophyte sub-species variants occur in subtropical and tropical oceans

At BATS, categorization of sub-species variants along the spectrum from persistent to ephemeral across stability periods showed that a handful of persistent ASVs had the greatest contributions to eukaryotic phytoplankton in terms of relative abundance (Fig. 6). We compared these findings to data from the tropics[53,59] as well as higher latitudes[19,23] for sub-species exhibiting >10% of plastid amplicons in at least one sample (Table 1), and for other notable sub-species (Supplementary Data S4). In the tropical Bay of Bengal, the prominent DCM has higher nutrient concentrations than the surface, akin to BATS, however the vertical structure is driven by a strong salinity gradient, not temperature. Class II taxonomic composition had similarities to BATS and the dominant BATS sub-species variants were present except *M.* cand. species 1. For example, *M. commoda ss* ASV345 contributed to plastid amplicons in the surface 1-2 m in the Bay of Bengal, especially in southern stations[59]. *Ostreococcus* Clade OII contributed greatly at the surface in southern stations and dominated the DCM throughout, where *Bathycoccus calidus* was also observed. Although not among dominants, Class VI *Prasinoderma* was also detected with a sub-species variant identical to one from the SS top 40 m.

In the Southern Caribbean, temperatures were similar to BATS SS surface water, but nutrient concentrations were higher, especially above coral reefs and in mangroves[53]. A species absent from BATS, *Ostreococcus bengalensis*, was present in mangroves as well as the southern Bay of Bengal stations[59]. Otherwise, similar species and sub-variants were observed, with *M. commoda ss* as the dominant, including the key BATS sub-species variant (Table 1, Supplementary Data S4). However, unlike at BATS, *M. bravo* reached high relative abundances (a different sub-species variant than at BATS where this species was low in abundance) and alongside *M.* candidate species 1 and 2, formed the major *Micromonas* species[53], all of which had ASVs that matched those found at BATS. *Ostreococcus* OII was the major *Ostreococcus* above coral reefs and in the open sea. *Bathycoccus calidus* had similar distributions as *Ostreococcus* OII, although lower relative abundances[59]. In these marine environments *B. prasinos* was rarely detected, establishing *B. calidus* as the dominant *Bathycoccus* in warm oligotrophic environments[25,47]. Overall, these results indicate that abundant ASVs from different prasinophyte species in the subtropics are globally distributed in warm oceanic waters and, at least at BATS, are persistently present.

At higher latitudes, large picoprasinophyte contributions to phytoplankton biomass have also been observed[23]. In the western North Atlantic spring phytoplankton bloom Class II is highly abundant and dominates among prasinophytes, but *Ostreococcus* Clade OI, which is often reported in mesotrophic and coastal waters[17,25], is the main contributor in the subtropically-influenced spring community[23]. *Ostreococcus* Clade OII increased in relative abundance moving southward. In subtropical winter and subpolar samples from the western North Atlantic[23] as well as the subpolar eastern North Atlantic[19], *M. commoda*-like Clade C (*sensu* Simmons/Šlapeta) and *Micromonas polaris* (Clade E2) dominate relative abundances (plastid-derived) in many samples. While *M.*

*commoda*-like Clade C and *M. polaris* were not detected at BATS, other prominent sub-species variants of *M. pusilla* and *M. bravo* were seen (albeit rarely) as well as *M.* candidate species 1 ASV81.

Diversity, genomic differentiation, and modelled temperature tolerances have led to *Micromonas* being proposed as sentinels of ocean change[60,61]. Our field data from BATS, other studies, and from higher latitudes demonstrate that while *M. commoda-like* Clade C and *M. polaris* are effectively not present in the subtropics or tropics, *M. bravo* and *M.* candidate species 1 can extend northwards in water masses with subtropical influences. Dominants for subtropical and tropical waters are *M. commoda ss* and *M.* candidate species 1. Moreover, specific sub-species are present in multiple subtropical and tropical settings. The distributions of dominant sub-species variants are linked to environmental conditions and stratification structures that are predicted to change with ocean warming – thus the mapping accomplished herein provides a baseline against which such change can be assessed. These findings propel the use of the *Micromonas* genus as indicator species of ocean conditions and climate change[60].

## Reservoirs of diversity and genomic potential

Two types of water column reservoirs were observed. The DCM acted as one – being the summer home to spring bloom dominants. These results expand upon the hypothesis put forward by Treusch et al.[17] that the same groups (in that case genera) that comprise the spring bloom relocate to the DCM as the water column stratifies and stabilizes. Indeed, the dominant prasinophyte species, and ASVs, found throughout ML during the DM, specifically *B. calidus* ASV177, *M. commoda ss* ASV61, *M.* candidate species 1 ASV81, and *Ostreococcus* OII ASV6 and ASV77, were all observed within and below the DCM during the SS stability period (Fig. 4). Importantly, the persistence of relocated Class II dominants within the DCM in the SS and AT stability periods indicated they were poised to increase cell abundance upon DM period convective mixing, likely underpinning their spring bloom importance.

Despite the persistence of key sub-species variants in the SS stability period DCM (Fig. 4), there are both significant and subtle differences in the prasinophyte community composition when comparing ASVs from the entire ML during the DM period to the DCM and other stability periods. These differences could reflect temporal and depth partitioning. For example, the vertical distribution patterns of *M.* candidate species 2 ASV1156 as well as *M. commoda ss* ASV273 in summer and winter may indicate a wide thermal tolerance or affinity for co-associated factors (Fig. S6b). Other prasinophytes, even other Class II ASVs such as *Ostreococcus* OII ASV33032, were only detected during summer, indicating adaption to SS DCM conditions (see also below). Thus, while the DCM acts as a reservoir to key players from the bloom period, there is also a broader reservoir of novel genetic diversity in the SS stability period.

The second type of reservoir of prasinophyte genetic potential identified at BATS may have special importance to changing ocean conditions. We observed the greatest diversity of prasinophytes in the SS stability period, based on overall ASVs assigned to prasinophytes (Fig. 6a,b). These SS

community members exhibited little overlap between years, and were generally much lower in relative abundance than persistent sub-species variants (Fig. 6c, d). Moreover, compared to ephemerals from other stability periods, the numbers of SS ephemerals also varied more from year to year based on our sampling and detection limits. SS ephemeral ASVs often displayed depth zonation patterns, including around the DCM where 'persistents' had much higher relative abundance (Fig. S5b). Even when separating the SS upper ML from SS DCM data, the number of ephemeral ASVs for each was higher than in other stability periods (Fig. S5). It is important here to recognize that possible functional redundancy based on photosynthetic capabilities does not capture the genomic differences that shape growth optima and requirements, the key physiological aspects to when and where a species or sub-species variant may thrive. The diverse sub-species observed in summertime reflect a reservoir of genetic diversity capable of responding (differentially) to shifting environmental conditions and potentially leading to episodic importance in the phytoplankton community.

## Stratification dynamics influence the export of picoprasinophytes

One of the complicating factors for understanding the fate of picophytoplankton is that the cells presumably do not sink on their own accord because of their small size[62]. A growing body of evidence backs export of picophytoplankton through other mechanisms such as eddy-driven subduction at higher latitudes[63] and increasing particle density as a result of aggregation via exopolymeric substances or repackaging in fecal pellets[62]. Indeed, molecular approaches demonstrate that picoprasinophytes have been exported to depth in Arctic waters and in the North Atlantic bloom through different physical mechanisms[19,23]. At BATS, *Ostreococcus*, *Micromonas*, and *Bathycoccus* have been reported in sediment traps at 150 m (2008-2010)[64]. Furthermore, a microscopy-based sediment trap particle analysis from 150-300 m (2017-2018) reported *Bathycoccus* (which, unlike *Ostreococcus* and *Micromonas*, does have distinguishing scales) in phytodetrital and fecal aggregates, particularly in spring[10].

In the present study, the deepest mixing observed occurred in April 2017 with the ML extending to >300 m, well below the euphotic zone (Fig. 1). While the 2017 DM appeared to have deeper convective mixing than the other years sampled, this observation is likely due to the 2017 sampling serendipitously coinciding with a deep mixing event that was not captured during the DM period of 2018-2019. This higher-resolution targeted sampling of the event clearly captured the shoaling of the ML as the water column warmed and stratified and appeared to trap prasinophytes at depth. Thus, the DM prasinophyte community—mostly comprised of *M. commoda ss*[33], *Micromonas* candidate sp. 1, *B. calidus*, and *Ostreococcus* Clade OII throughout the mixed layer—not only would be subject to losses by predation or viral lysis but additionally could be exported or, at minimum, the biomass would have become a resource for communities below the euphotic zone. This type of physical mixing and subsequent trapping of photoautotrophic biomass phenomenon has been reported in other ocean regions[65,66], and our studies indicate it serves as a mechanism for picophytoplankton export.

## Conclusions and perspectives

Our studies establish the importance of prasinophytes, particularly picoplanktonic Class II, to open ocean phytoplankton communities and as contributors to bloom periods. By leveraging vertically-resolved data across the seasonal cycle, we demonstrate the dominance of specific species and sub-species variants – and coherent distributional patterns across periods of thermal stability and mixing throughout the euphotic zone. The dominant Class II sub-species variants observed during DM tracked the nitricline as the system stratified at the BATS site. These results confirm and expand upon the hypothesis that dominant DM prasinophytes relocate to the DCM during the stratified period[17]. Moreover, specific sub-species variants (ASVs) from the major prasinophytes (*Ostreococcus* Clade OII, *M. commoda ss*, and *B. calidus*) in the DM and DCM persevered across stability periods. Thus, these taxa were poised to use nutrient pulses associated with convective mixing at BATS, enabling strong recolonization of the entire ML during the periods when primary production has been shown to be highest. Remarkably, these dominant sub-species variants are also present at high relative abundances in the tropical Indian Ocean (Bay of Bengal) and tropical North Atlantic (Caribbean Sea) (Table 1).

Other sub-species variants from Class II and other prasinophyte classes were often ephemeral but provide a reservoir of diversity associated with different stability periods, particularly summer waters — a period considered akin to open ocean desertification scenarios wherein stratification intensifies as the water column warms. These more ephemeral sub-species variants do not appear to be entrained or involved in the same mechanisms as those that shift from a DM distribution throughout the ML into the DCM, but rather have distinct patterns or punctuated appearances. The overall degree of heterogeneity observed at BATS appears to reflect a pool of genetic variants that utilize different ecological strategies than today's persistent variants and are uniquely optimized for life in a stratified warm water ocean.

Importantly, while mechanisms of export for picophytoplankton are an area of active research[19,62] given their low propensity of sinking, we find that as the surface layer stratifies, the DM/ST primary producer communities can become 'trapped' under the ML, in a manner that results in vertical export from the euphotic zone. The organic matter captured in the redistribution of picoprasinophytes from the surface into the mesopelagic resulting from convective mixing at BATS would be included in estimates of suspended organic carbon export[67] at a magnitude that remains to be quantified.

We demonstrate that prasinophytes are important to phytoplankton community structure, biomass, and likely export flux in oligotrophic ecosystems. Moreover, we identify an underlying mechanism that drives seasonal distributions and blooming of prasinophytes. These pivotal findings rely on vertically resolved data and contextualization within the annual cycle – allowing tracking of sentinel species, improvement of predictive NPP models, and understanding of its ecosystem fate. Such advancements are essential for predicting phytoplankton community transitions associated with ocean warming.

## Methods
### Oceanographic sampling
Seawater samples for DNA were collected at 8 depths between 1 and 300 m using Niskin bottles affixed to a conductivity-temperature-depth (CTD) profiling rosette from 2018 to December 2019, and analyzed alongside data from profiles collected under the same umbrella project from July 2016 to the end of 2017[68] totalling 79 profiles and 431 samples. Fifty-six profiles were within 5 km of the BATS station (31°40' N, 64°10' W), while the remaining 23 were within 6 to 109 km (Fig. 1a). Four L of seawater were filtered through 0.22 μm Sterivex™ (Millipore) filters for DNA as in[68]. This type of sampling and that for nutrients and pigments was conducted ~monthly, with additional dedicated BIOS-SCOPE process cruises in September 2016, April 2017, July 2018, and July 2019. The process cruises that provided more resolved sampling effort, e.g., daily profiles and sampling was performed synoptically (Supplementary Data S1).

### Physico-chemical analyses
The surface mixed layer depth (MLD) was defined as the depth where density was ≥ surface sigma-t plus 0.125 kg m$^{-3}$. DCM determinations were made using in vivo fluorescence. Specifically, we determined the region where measured fluorescence spanned ± 35% of the maximum fluorescence (so inclusive of the maximum and spanning depths on either side). This region was termed the DCM for the periods where the MLD was shallower than the DCM region. The water column stability periods at BATS were then defined relative to the DCM and MLD[32], with the DM defined as when the MLD is greater than the euphotic zone depth (defined as 0.1% of surface photosynthetically active radiation, PAR), the ST period beginning when the MLD shoals to ~100 m, the SS period beginning when

the MLD shoals above the DCM and remaining that way until the first entrainment of the DCM into the mixed layer (ML) marks the beginning of the AT. The 1% irradiance level at 80–120 m during the DM was used as a comparison point for the DCM, which was usually at similar depths.

Concentrations of Chl *a*, nitrate+nitrite, phosphate, and POC were determined at 12-13 depths per profile and measured according to prior methods[5]. Nitrate+nitrite and phosphate below the limits of detection (0.05 µmol and 0.03 µmol, respectively) were reported as zero.

## DNA extraction and amplicon sequencing

DNA was extracted from using a phenol chloroform protocol and the V1-V2 region of the 16S rRNA gene then PCR amplified using the primers 27F (AGAGTTTGATCNTGGCTCAG) and 338RPL (GCWGCCWCC CGTAGGWGT) with 'general' Illumina overhang adapters[69]. Amplicon libraries were pooled in equimolar concentrations prior to 2×250 paired-end (MiSeq Reagent Kit v2) sequencing. On average 60,279 ± 26,939 V1-V2 amplicons were sequenced per sample. Sequences were not available for March 2017 at the time of this study. Sequence data were trimmed [--p-trunc-len-f 180 --p-trunc-len-r 150], dereplicated, checked for chimeras, and assigned to amplicon sequence variants (ASVs) using the DADA2 R package v1.14.0[70].

## Phylogenetic analyses

Quality controlled V1-V2 16S rRNA gene ASVs were classified using best node placement in a rewritten Python3 version (source code at github https://github.com/BIOS-SCOPE/PhyloAssigner_python_UCSB) of the phylogenetic placement method PhyloAssigner, see[69], which relies on maximum likelihood approaches and 16S rRNA gene reference trees/ alignments. ASVs were first placed on a global 16S rRNA gene reference tree; those assigned to plastid or cyanobacterial nodes were subsequently placed on a more resolved plastid and cyanobacteria tree[15]. Amplicons assigned to the Viridiplantae (chlorophytes, prasinophytes, and other green lineage organisms) in this second classification step were placed on a phylogenetic reconstruction of prasinophyte sequences (requirement: length >1200 bp and inclusive of the V1-V2 hypervariable region which is absent from some shorter GenBank entries) from GenBank nr and MMETSP[71] that represented all classes (including prasinodermophytes). Because plastids from chlorarachniophytes, euglenids, and dinophytes contain plastids acquired from green algae, representative sequences from these groups were also incorporated. Nine streptophyte sequences were used as outgroups. The final 149 sequences were aligned using MAFFT, and regions of unambiguous alignment were identified using MUST and removed, as were all gap-containing positions[72,73]. The final alignment consisted of 1080 characters. A best-fit model of nucleotide evolution (GTR + Γ+I) was determined using the likelihood ratio test implemented with jModeltest[74]. The Maximum Likelihood phylogenetic analysis was calculated using PhyML[75] and 1000 bootstrap replicates. The reconstruction resolved most classes and delineated several genera to the species-level contingent on the availability of reference sequences (Fig. S2). Within a cluster grouping *Micromonas*, *Mamiella*, and *Mantoniella*, the near full-length 16S did not discriminate among *Micromonas* Clades A and B, but did between A-B versus C, D, and E *sensu*[27,33]. Therefore, sequences assigned to these groups were manually interrogated to provide greater resolution.

Placement accuracy for V1-V2 16S amplicons was tested by building a Maximum Likelihood reconstruction using the same Viridiplantae reference alignment trimmed to the size of our amplicons. Without masking, the tree was then inferred using the same methods as above, which discriminated the same clades as the near-full-length 16S phylogenetic analysis. For *Micromonas* ASVs requiring additional resolution, 16S rRNA ASVs were compared to 18S rRNA ASVs from a study that had a subset of the same samples[32]. Additionally, trees were constructed using V1-V2 16S and V4 18S rRNA amplicons from *Micromonas* (and 3 *Mantoniella* as outgroup) to connect clade information between 16S and 18S rRNA gene data (Fig. S3; see also Statistical analyses). Other prasinophyte V1-V2 16S ASVs with a last common ancestor (LCA) best node mismatch and low nucleotide

% identity to their assigned group were used as queries against NCBI nr, and iterative searches were performed with returned NCBI sequences and the PR[2] database[76].

## Statistical analyses

Temperature, CTD-derived fluorescence, and salinity data from continuous CTD measurements as well as Chl *a*, nitrate+nitrite, and phosphate concentrations were averaged by stability period and depth[77]. The normality of the metadata distributions by depth and water column stability period was tested using the Shapiro test; non-normally distributed data were compared using the Kruskal-Wallis and Dunn tests using the R stats and rstatix packages[78].

Rarefaction curves for all ASVs and for plastid ASVs only were generated using vegan and the rareslope function characterized the level of sequence saturation[79]. Ultimately 431 samples with final rarefaction curve slopes indicating saturation (slope<0.1) were used for further analysis: 73 were from the surface (~1–5 m), 63 from the DCM, and the remaining 295 were from 20–300 m (Supplementary Data S1). All samples from 1–200 m had ≥50 plastid amplicons and had rarefaction slopes indicating saturation for plastid amplicons; the 10 samples maintained in our study that did not meet these criteria were from ≥250 m (Fig. S8, S9). Samples were normalized relative to total amplicon abundance per sample, relative to plastid amplicons per sample, and relative to prasinophyte (with Class VI included) amplicons per sample. Averages and standard deviations of prasinophyte groups were computed according to depth and stability period. Percent contributions by depth and stability period were tested for normality and compared as for the environmental data comparisons. Environmental data (Supplementary Data S1) was used synoptically.

The Spearman correlation was used to compare POC to the CTD-derived Chl fluorescence signal ($n = 1021$, with a February 2017 profile removed due to erratic signals) and Chl *a* ($n = 446$) since the data was not normally distributed. Spearman correlations were also used to compare prasinophyte amplicon contribution to the CTD-derived Chl fluorescence signal ($n = 425$ for all depths) and Chl *a* ($n = 409$ for depths ≥250 m). The relationship between Chl *a* and prasinophyte contributions to plastid amplicons was explored as quadrants (above and below 0.188 µg kg$^{-1}$, the 75th percentile Chl *a* value in the surface 140 m from June 2016 to December 2019, and 33.2%, the 75th percentile of prasinophyte contributions to plastid amplicons).

To determine assignments not resolved using ASV phylogenetic placement, the V1-V2 16S rRNA and V4 18S rRNA ASVs from a subset of samples ($n = 207$) sequenced previously[32] were compared using the Spearman correlation with the cor.test function and linear regression in R. Samples in which unplaced *Micromonas* ASV81 and ASV1156 and/or candidate species 1[34] or 2[26] defined by 18S were detected in at least one of these two markers were compared, and the comparison input was relative abundances out of all *Micromonas* in each sample, and refined to the subset of samples with ≥10 *Micromonas* amplicons ($n = 30$ for ASV81/ *Micromonas* candidate species 1 and $n = 14$ for ASV1156/ *Micromonas* candidate species 2). Contributions of these 16S ASVs were plotted against contributions of the candidate species to 18S *Micromonas* amplicons. V1-V2 16S ASVs assigned as *Micromonas commoda* Clade A were also compared to 18S ASVs assigned to that clade because both marker gene sequences are known for this species. Additionally, V1-V2 16S ASVs grouped as a putative Class IX were compared using the correlation approach to V4 18S ASVs for sequences in[32] identified in PR[2] as prasinophyte Clade 9.

To investigate potential differences in the overall prasinophyte community with respect to stability periods and environmental conditions, a partial canonical correspondence analysis (pCCA) was performed on Hellinger-transformed abundance data of prasinophyte ASVs using the vegan package in R. The stability period served as the constraining variable to control for effects of water column stability on the (abiotic) environmental variables temperature, salinity, nitrate+nitrite, phosphate, and depth[79] and a second analysis was performed with Chl *a* data as well. The significance of associations was determined using the ANOVA-like permutation test for pCCA, anova.cca. Analysis of similarity (ANOSIM) tests

were also performed on Hellinger-transformed abundance data of prasinophyte ASVs to compare across years, depths, and stability periods. To determine if there was a uniform prasinophyte community throughout the ML during DM, the ML DM prasinophyte community was compared across depths ($n = 70$). The ML DM community was also compared to that of the stratified surface ($n = 45$) and DCM samples ($n = 47$) across years. ASVs were characterized from ephemeral to persistent based on the number of stability periods they were detected in between 2017-2020.

## Visualization

Most figures and data explorations were performed using ggplot2, and heatmaps were generated using the heatmap.2 function in the gplots package in R[80]. Relative abundance of major ASVs out of total 16S amplicons was plotted across depths using the mba package for Multilevel B-Spline Approximation in R[77]. Boxplots were generated using ggplot2 and the tidyverse, readxl, lubridate, and patchwork packages in R[81,82]. Venn diagrams, pie charts, UpSet plots, and a barplot further characterizing these distributions were plotted using ggplot2.

## Reporting summary

Further information on research design is available in the Nature Portfolio Reporting Summary linked to this article.

## Data availability

Oceanographic and environmental data (temperature, salinity, CTD-derived Chl fluorescence, Chl *a*, nitrate+nitrite, phosphate) can be found in the BCO-DMO repository[83,84] (https://www.bco-dmo.org/dataset/861266 and http://lod.bco-dmo.org/id/dataset/3782) and at the BATS website (https://bats.bios.asu.edu/bats-data/) as well as in supplementary date files for data from a narrower set of information used in this manuscript. V1-V2 16S rRNA gene amplicons data can be found in the NCBI SRA repository under the project number PRJNA769790.

## Code availability

The source code for the phylogenetic placement method PhyloAssigner can be found at github (https://github.com/BIOS-SCOPE/PhyloAssigner_python_UCSB). The code used for statistical analyses and figure generation can be found at (https://github.com/Eckmannspiral/BATS-prasinophyte-paper).

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

## Acknowledgements

We thank the captains and crews of the RV *Atlantic Explorer* and RV *Endeavor*, the BATS Program and staff (funded through NSF-OCE 1258622 and 1823636), BIOS-SCOPE technicians and interns, and Valeria Jimenez for comments. This work was supported by Simons Foundation International's BIOS-SCOPE Program and National Science Foundation DEB-1639033 (to AZW). Finally, we thank the reviewers for constructive criticism.

## Author contributions

C.A.E., A.Z.W., C.A.C., S.J.G. and R.J. conceived and coordinated study. C.B., F.W., C.A.C., S.J.G., L.B-B., K.L.V., R.J.P. and L.M.B. collected samples or performed laboratory work. C.B. performed phylogenetic reconstructions. K.L.V. and F.W. performed bioinformatics. C.A.E. performed all statistical analyses with support from J.S. and F.W. C.A.E. and A.Z.W. formulated analyses with conceptual input from R.M.K. and C.A.C. C.A.E and A.Z.W. wrote the manuscript with major input from R.M.K., C.B., and C.A.C. All authors read and approved the manuscript.

## Competing interests

The authors declare no competing interests.
