## [Peer review File · Communications Earth & Environment]

Web links to the author's journal account have been redacted from the decision letters as indicated to maintain confidentiality.

8th Feb 24

Decision letter and referee reports: First Round

Dear Professor Worden,

Your manuscript titled "Deconvolution of seasonality exposes dominant species and niche partitioning strategies of open ocean picoeukaryotic algae" has now been seen by 2 reviewers, and we include their comments at the end of this message. They find your work of interest, but some important points are raised. We are interested in the possibility of publishing your study in Communications Earth & Environment, but would like to consider your responses to these concerns and assess a revised manuscript before we make a final decision on publication.

We therefore invite you to revise and resubmit your manuscript, along with a point-by-point response that takes into account the points raised. In particular, we ask that you consider the points raised by Reviewer #1 regarding salinity and chlorophyll, improve the clarity of figure 6, and address the point raised by Reviewer #2 regarding primary production versus biomass and appropriately adjust your claims. Please highlight all changes in the manuscript text file.

Please use the following link to submit your revised manuscript, point-by-point response to the referees' comments (which should be in a separate document to any cover letter), a tracked-changes version of the manuscript (as a PDF file) and the completed checklist:

Link Redacted

We hope to receive your revised paper within six weeks; please let us know if you aren't able to submit it within this time so that we can discuss how best to proceed. If we don't hear from you, and the revision process takes significantly longer, we may close your file. In this event, we will still be happy to reconsider your paper at a later date, as long as nothing similar has been accepted for publication at Communications Earth & Environment or published elsewhere in the meantime.

Please do not hesitate to contact us if you have any questions or would like to discuss these revisions further. We look forward to seeing the revised manuscript and thank you for the opportunity to review your work.

Best regards,

Clare Davis, PhD

Senior Editor

Communications Earth & Environment

www.nature.com/commsenv/

@CommsEarth

EDITORIAL POLICIES AND FORMATTING

Editorial Policy: Policy requirements (Download the link to your computer as a PDF.)

Furthermore, please align your manuscript with our format requirements, which are summarized on the following checklist:

Communications Earth & Environment formatting checklist

and also in our style and formatting guide Communications Earth & Environment formatting guide .

*** DATA: Communications Earth & Environment endorses the principles of the Enabling FAIR data project (<http://www.copdess.org/enabling-fair-data-project/>). We ask authors to make the data that support their conclusions available in permanent, publically accessible data repositories. (Please contact the editor if you are unable to make your data available).

All Communications Earth & Environment manuscripts must include a section titled "Data Availability" at the end of the Methods section or main text (if no Methods). More information on this policy, is available at <http://www.nature.com/authors/policies/data/data-availability-statements-data-citations.pdf>.

If a community resource is unavailable, data can be submitted to generalist repositories such as figshare or Dryad Digital Repository. Please provide a unique identifier for the data (for example a DOI or a permanent URL) in the data availability statement, if possible. If the repository does not provide identifiers, we encourage authors to supply the search terms that will return the data. For data that have been obtained from publically available sources, please provide a URL and the specific data product name in the data availability statement. Data with a DOI should be further cited in the methods reference section.

REVIEWER COMMENTS:

Reviewer #1 (Remarks to the Author):

This manuscript is a comprehensive overview of the seasonal and vertical distributions of picoeukaryotic algae, prasinophytes, in the North Atlantic Subtropical Gyre at BATS across three years. Prasinophyte

diversity from class, species and amplicon sequence variant (ASV) level using 16S rRNA gene amplicons were identified and categorized in terms of how they partitioned the time and space (primarily depth) niches. Rather than stick with date-based seasons, the authors defined “stability periods” which roughly correspond to Spring, Summer, Autumn and Winter but provide a more realistic delineation of physical properties that correspond to likely ecological factors influencing distributions. The authors have been able to identify dominant, persistent and ephemeral species for the different stability periods, presented a nice statistical method for identifying ASVs from uncultured species (section at Line 177), and made headway into understanding the heterogeneity of the prasinophytes in this region. They also are candid about the limitations of their method (line 199). The level of detail of this study and its clear presentation will serve as an important reference in the field of marine microbial oceanography for a major component of the picophytoplankton consortium.

Overall, the manuscript is very well written and the presentation of the data is thorough. I see no major issues with it. I do have a few points that need to be addressed by the authors along with some minor corrections that I outline below.

1. The authors introduce the term “stability period” (line 131) to avoid associating these physical events with specific seasonal time periods: ST = Spring Transition, SS = Stratified Summer, AT = Autumn Transition, and DM = annual Deep Mixing, corresponding roughly to Spring, Summer, Autumn, and Winter, respectively. This way of representing the seasonal periods works quite well as the timing does not always coincide with human calendar periods. However, it can be a bit hard to parse out the abbreviations amongst some of the others, such as mixed layer (ML) and deep chlorophyll maximum (DCM), e.g. line 542 “...from the entire DM ML to the DCM and other periods.”. I suggest providing a table of abbreviations to make it easier for readers.

Also, there is some inconsistency with the use of the word “season” and “stability period” and their representative terms in various locations throughout the manuscript, e.g. titles at line 205 and 425, lines 536-539, and Fig. 6 and its legend. I suggest that the authors consider going through the manuscript to ensure the consistent use, and specifically fix Fig 6 and/or its legend.

2. Salinity is mentioned as not manifesting pronounced seasonal patterns like temperature (line 148), but then it was found to be significantly associated with stability periods in the CCA (line 302 and Fig. 5). Please clarify what you mean in line 148 as it seems inconsistent with the results presented in Fig 5. It also would be nice to see salinity presented in Fig. 1b if it is not too difficult to add.

3. Chl a was found to be significantly connected to stability periods (Fig 1c) and prasinophyte ASV distributions (Fig 3) yet was not used in the coordination analyses (Fig 5). Is there a reason for this?

4. Fig. 6 is extremely challenging, particularly Fig. 6b which is split with year 2017 to the left and year 2018 and 2019 on the right. All three years really need to be aligned vertically with each other. Having the colors associated with the prasinophyte classes/species below year 2017 almost looks like the different classes/species corresponds to the Upset plot columns, so the prasinophyte “color legend” needs to be elsewhere in the figure. One possible solution that might work would be to put Fig. 6c to the right of Fig 6b (with the Upset plots for the 3 yrs going from 2017 down to 2019) and the color legend rotated 90deg and located to the right of Fig 6c. [Not sure if this is clear!]

Additionally, it is not clear what 2, 3 and 4 periods mean in Fig 6a. Perhaps I missed it in the text. Please explain.

Other minor comments:

Line 128 – define ASV

Line 305 – need a comma between ASV1156 and Mamiella

Line 435 – replace “as” with “are”?

Line 467 – missing rRNA after 18S

Line 513 – Should “Will” be changed to “While”?

Line 608 – clarify which dominant sub-species variants are also in high abundances elsewhere and provide a reference.

Reviewer #2 (Remarks to the Author):

The manuscript by Eckmann et al. is well written with in-depth discussion. This study, conducted at the Bermuda Atlantic Time-Series Study site (BATS), focuses on a biogeochemically significant oligotrophic region. It presents a solid dataset spanning three years, utilizing 16S chloroplast metabarcoding to study the eukaryotic phytoplankton dynamics at a fine taxonomic level, from the surface ocean to 300m. One major finding of the study is the prevalence of Class II prasinophytes during spring blooms and at the Deep Chlorophyll Maximum (DCM) when water column is well stratified. I believe the results of this

study are deserving of publication in *Communications Earth & Environment*, as they hold significant value for researchers in biological oceanography, biogeochemistry, and microbial ecology.

Some minor points. Two general issues:

The study presents community structure as a percentage of the total community chloroplast 16S reads. At the community level, the potential biases of the method should be further acknowledged, including variations in chloroplast DNA (cpDNA) copy numbers per cell, and the influence of environmental factors and growth phase on cpDNA copy numbers in certain organisms. Koumandou and Howe 2007, The copy number of chloroplast gene minicircles changes dramatically with growth phase in the dinoflagellate *Amphidinium operculatum*. *Protist*, 158(1), pp.89-103.

Without direct group specific rate measurements, should tune down a bit on the conclusion that prasinophytes are important to primary production. Based on the data presented in this study, it's more appropriate to state that they are important to the biomass/standing stock. For example, at DCM, despite high biomass the photosynthesis rates may be low due to light limitation.

Detailed:

Line 282: don't see *M. commode* in Figure 4a. Please change or clarify.

Figure 5: CCA plot. Add % variation explained for CCA1 and CCA2 axes

Line 311 – 319 and Figure 6: Would sequencing depth for each sample influence unique or rare ASVs identified in each pool? Were these resampled at an even depth for comparison?

Line 381: provide R^2 and p-value for Chl vs. POC

Line 475: change 'sub-subspecies' to 'sub-species'

Figure S2. add p-value

Author responses: First round

We are grateful for the thoughtful reviews and comments and feel that the manuscript has
been improved while addressing these comments. We have added two analyses to address
comments by Reviewer 2 (new Fig. S4, rarefaction for plastid amplicons, previously only
rarefaction of all amplicons was shown) and Reviewer 1 (new Fig. S9, CCA with abiotic
factors and Chl a data added per the reviewer query, previously only the abiotic factors were
investigated).

Also please note that to accommodate a definition of the stability periods/seasonality in the
abstract (as brought up by the reviewer) we added a sentence to the abstract which makes it
longer than 200 words. For the online form we put in the old abstract and hope that this is
ok (since the online requires 200 words).

**REVIEWER COMMENTS:**

**Reviewer #1 (Remarks to the Author):**

*This manuscript is a comprehensive overview of the seasonal and vertical distributions of*
*picoeukaryotic algae, prasinophytes, in the North Atlantic Subtropical Gyre at BATS across*
*three years. Prasinophyte diversity from class, species and amplicon sequence variant (ASV)*
*level using 16S rRNA gene amplicons were identified and categorized in terms of how they*
*partitioned the time and space (primarily depth) niches. Rather than stick with date-based*
*seasons, the authors defined "stability periods" which roughly correspond to Spring, Summer,*
*Autumn and Winter but provide a more realistic delineation of physical properties that*
*correspond to likely ecological factors influencing distributions. The authors have been able to*
*identify dominant, persistent and ephemeral species for the different stability periods,*
*presented a nice statistical method for identifying ASVs from uncultured species (section at*
*Line 177), and made headway into understanding the heterogeneity of the prasinophytes in*
*this region. They also are candid about the limitations of their method (line 199). The level of*
*detail of this study and its clear presentation will serve as an important reference in the field of*
*marine microbial oceanography for a major component of the picophytoplankton consortium.*

*Overall, the manuscript is very well written and the presentation of the data is thorough. I see*
*no major issues with it. I do have a few points that need to be addressed by the authors along*
*with some minor corrections that I outline below.*

Thank you very much for these positive comments; we have tried to accommodate
all suggestions.

1. The authors introduce the term "stability period" (line 131) to avoid associating these

*physical events with specific seasonal time periods: ST = Spring Transition, SS = Stratified*
*Summer, AT = Autumn Transition, and DM = annual Deep Mixing, corresponding roughly to*
*Spring, Summer, Autumn, and Winter, respectively. This way of representing the seasonal*
*periods works quite well as the timing does not always coincide with human calendar periods.*
*However, it can be a bit hard to parse out the abbreviations amongst some of the others, such*
*as mixed layer (ML) and deep chlorophyll maximum (DCM), e.g. line 542 "...from the entire*
*DM ML to the DCM and other periods." I suggest providing a table of abbreviations to make it*
*easier for readers.*

We have now added these abbreviations and their definitions at the top of Fig. 1
panel B; thank you for noticing this. We also changed line 542 to: "...from the entire ML
during the DM period to the DCM and other stability periods." to try to make this less
difficult to follow as suggested above.

*Also, there is some inconsistency with the use of the word "season" and "stability period" and*
*their representative terms in various locations throughout the manuscript, e.g. titles at line 205*
*and 425, lines 536-539, and Fig. 6 and its legend. I suggest that the authors consider going*
*through the manuscript to ensure the consistent use, and specifically fix Fig 6 and/or its*
*legend.*

Thank you for pointing out this inconsistency; we changed all instances of "season"
to "stability period." In the abstract we have added the sentence:

"Water column vertical structure was then used to delineate seasonal stability
periods more ecologically relevant than seasons defined by classical calendar dates."
ensuring that usage of e.g. winter etc. would still work.

We have continued to use the words seasonal or seasonally, but again have changed
all instances of season to stability period (alongside noting that we are redefining seasons
according to water column structure). To avoid further confusion we also amended a
sentence in the final paragraph of the introduction to state:

"Rather than working with set seasons (defined by calendar date), we identified
varying periods of water column stability (hereafter stability periods) that reflected
seasonality within the water mass:..."

*2. Salinity is mentioned as not manifesting pronounced seasonal patterns like temperature*
*(line 148), but then it was found to be significantly associated with stability periods in the CCA*
*(line 302 and Fig. 5). Please clarify what you mean in line 148 as it seems inconsistent with the*
*results presented in Fig 5. It also would be nice to see salinity presented in Fig. 1b if it is not*
*too difficult to add.*

Thank you for pointing this out, to clarify we have now rewritten the point that was
at line 148 to now read:

"Temperature variation in the euphotic zone demonstrated seasonal thermal
stratification (Fig. 1b). Euphotic zone salinity patterns were less regular, with longer
periods of lower salinity during the SS and AT but also shorter low-salinity periods
during the DM (Fig. S4a)."

Due to the size of Fig. 1 in its current state we have left this as a supplemental figure –
however if the reviewer and editor feel it is essential to have it in the main figure, we are
happy to make that change.

*3. Chl a was found to be significantly connected to stability periods (Fig 1c) and prasinophyte*
*ASV distributions (Fig 3) yet was not used in the coordination analyses (Fig 5). Is there a*
*reason for this?*

The coordination analysis was based on abiotic factors – and because Chl *a* is a
property of the prasinophytes themselves (i.e., they contribute to the Chl *a* signal) we did
not include Chl *a* in the CCA (again as it is not necessarily an external factor). However, we
have now performed this analysis (a version of the CCA with Chl *a* added); Class II ASVs as a
whole grouped closer to the Chl *a* vector than many other prasinophyte groups (aligning
with expectations based on Fig. 3). Additionally, unlike salinity and temperature, Chl *a* was
not a significant variable in the CCA.

We have now modified this section and added a sentence to the main text stating:
"The overall ASV-level composition of the prasinophyte community across years, stability
periods, and depths was significantly associated with temperature and salinity variability
(permutation test for CCA using abiotic factors, $p < 0.001$; Fig. 5). A CCA incorporating Chl *a*
data (which at least in part would be derived from the prasinophyte algae) showed that
Class II grouped closer to the Chl *a* vector than many other prasinophyte groups, however,
unlike temperature and salinity, the association was not significant (Fig. S9)."

*4. Fig. 6 is extremely challenging, particularly Fig. 6b which is split with year 2017 to the left*
*and year 2018 and 2019 on the right. All three years really need to be aligned vertically with*
*each other. Having the colors associated with the prasinophyte classes/species below year*
*2017 almost looks like the different classes/species corresponds to the Upset plot columns, so*
*the prasinophyte "color legend" needs to be elsewhere in the figure. One possible solution that*
*might work would be to put Fig. 6c to the right of Fig 6b (with the Upset plots for the 3 yrs*
*going from 2017 down to 2019) and the color legend rotated 90deg and located to the right of*
*Fig 6c. [Not sure if this is clear!]*

*Additionally, it is not clear what 2, 3 and 4 periods mean in Fig 6a. Perhaps I missed it in the*
*text. Please explain.*

Thank you for these suggestions for simplifying this figure. We have now a.) aligned
the UpSet plots vertically and b.) then rotated and moved the prasinophyte color legend to
the right of the UpSet plots in an own box labeled "color legend for panels a and b". c.) We
also removed the colors of different years (to avoid redundancy with prasinophyte group
colors) and changed them to grayscale for Fig. 6a and Fig. 6c-d. We moved Fig. 6c-d to the
bottom of the figure. We also added this clarification (italicized) in the Fig. 6 legend:

"Relative abundance of prasinophyte sub-species variants out of plastid amplicons in
samples (symbols, also indicating depth) and stability periods (*i.e., detected in only*
*one or in multiple*) across years (light to dark grey) where these ASVs were found."

*Other minor comments:*

*Line 128 – define ASV*

Changed to: "We examined prasinophytes at the species and amplicon sequence
variant (ASV) levels..."

*Line 305 – need a comma between ASV1156 and Mamiella*

Added.

*Line 435 – replace "as" with "are"?*

Replaced.

*Line 467 – missing rRNA after 18S*

Added.

*Line 513 – Should "Will" be changed to "While"?*

Yes; now changed.

*Line 608 – clarify which dominant sub-species variants are also in high abundances elsewhere*
*and provide a reference.*

Thank you for pointing these out! We added a reference to Table 1 in line 608.

**Reviewer #2 (Remarks to the Author):**

*The manuscript by Eckmann et al. is well written with in-depth discussion. This study,*
*conducted at the Bermuda Atlantic Time-Series Study site (BATS), focuses on a*
*biogeochemically significant oligotrophic region. It presents a solid dataset spanning three*

146 years, utilizing 16S chloroplast metabarcoding to study the eukaryotic phytoplankton
dynamics at a fine taxonomic level, from the surface ocean to 300m. One major finding of the
study is the prevalence of Class II prasinophytes during spring blooms and at the Deep
Chlorophyll Maximum (DCM) when water column is well stratified. I believe the results of this
study are deserving of publication in *Communications Earth & Environment*, as they hold
significant value for researchers in biological oceanography, biogeochemistry, and microbial
ecology.

Thank you for the positive feedback; we tried to address all comments and have
added detailed information below on how this was accomplished.

*Some minor points. Two general issues:*

*The study presents community structure as a percentage of the total community chloroplast*
*16S reads. At the community level, the potential biases of the method should be further*
*acknowledged, including variations in chloroplast DNA (cpDNA) copy numbers per cell, and*
*the influence of environmental factors and growth phase on cpDNA copy numbers in certain*
*organisms. Koumandou and Howe 2007, The copy number of chloroplast gene minicircles*
*changes dramatically with growth phase in the dinoflagellate *Amphidinium operculatum*.*
*Protist, 158(1), pp.89-103.*

Thank you for bringing up this important line of thought. We are aware that
dinoflagellates have unique biology and plastids and have added a sentence that addresses
both this issue and the influence of genome replication state (growth phase) on rRNA copy
numbers. This states: "We note also that some taxa, such as dinoflagellates, have unique
plastids [42] that may not be effectively represented by the 16S rRNA PCR approach
employed here, and that for all organisms the state of genome replication and proximity to
division at the time of collection will influence numbers [31]."

Additionally, we have added another supporting reference that addresses this topic.

*Without direct group specific rate measurements, should tune down a bit on the conclusion*
*that prasinophytes are important to primary production. Based on the data presented in this*
*study, it's more appropriate to state that they are important to the biomass/standing stock. For*
*example, at DCM, despite high biomass the photosynthesis rates may be low due to light*
*limitation.*

Thank you, this is a good point. We changed the first sentence of the last paragraph
in the conclusion to now read:

"We demonstrate that prasinophytes are important to phytoplankton community
structure, biomass, and likely export flux in oligotrophic ecosystems."

Detailed:

Line 282: don't see M. commode in Figure 4a. Please change or clarify.

Thank you for catching this, it now correctly refers to Fig. 4d-e.

Figure 5: CCA plot. Add % variation explained for CCA1 and CCA2 axes

To address this the proportion explained by the constrained eigenvalues of CCA1 and CCA2 was added to the plot and indicated in the legend.

Line 311 – 319 and Figure 6: Would sequencing depth for each sample influence unique or rare ASVs identified in each pool? Were these resampled at an even depth for comparison?

Thank you for bringing this up. All samples from 1-140 m (the depths included in Fig. 6) were considered to be saturated based on rarefaction curve final slope thus it is unlikely that sequencing depth influenced the description as 'rare'. We added a figure (Fig. S4) where we plotted rarefaction curves for just the plastid amplicons.

In addition, just to make sure our text is not misinterpreted we have edited the sentence mentioning 'unique to a stability period' to now say:

"Forty-three of these were exclusive to the SS, which was the largest proportion of ephemeral ASVs detected only in one stability period (see Supplementary Text)."

Line 381: provide R² and p-value for Chl vs. POC

The Spearman ρ and p-value are stated in the results:

"Chl a and POC concentrations were positively correlated (Spearman $\rho = 0.497$, $p < 0.001$)."

Note that we did not square the Spearman ρ as that is not typically done for rank-based correlations since variation is less meaningful due to the transformation of data though ranking. Additionally, for balancing reasons we have not readded the stats values at this part of the discussion, since we have not restated all statistics throughout the discussion (since they are provided in the results). However, we can add them all to the discussion if the reviewer and editor feel it is essential.

Line 475: change 'sub-subspecies' to 'sub-species'

Fixed, thank you.

*Figure S2. add p-value*

We have added the p-value.

2nd Apr 24

Decision letter and referee reports: Second Round

Dear Professor Worden,

Your manuscript titled "Deconvolution of seasonality exposes dominant species and niche partitioning strategies of open ocean picoeukaryotic algae" has now been seen by our reviewers, whose comments appear below. In light of their advice we are delighted to say that we are happy, in principle, to publish a suitably revised version in Communications Earth & Environment under the open access CC BY license (Creative Commons Attribution v4.0 International License).

We therefore invite you to revise your paper one last time to address the remaining concerns of our reviewers. At the same time we ask that you edit your manuscript to comply with our format requirements and to maximise the accessibility and therefore the impact of your work.

EDITORIAL REQUESTS:

*****Please take care to match our formatting and policy requirements. We will check revised manuscript and return manuscripts that do not comply. Such requests will lead to delays. *****

SUBMISSION INFORMATION:

In order to accept your paper, we require the files listed at the end of the Editorial Requests Table; the list of required files is also available at <https://www.nature.com/documents/commsj-file-checklist.pdf> .

OPEN ACCESS:

Communications Earth & Environment is a fully open access journal. Articles are made freely accessible on publication under a CC BY license (Creative Commons Attribution 4.0 International License). This license allows maximum dissemination and re-use of open access materials and is preferred by many research funding bodies.

For further information about article processing charges, open access funding, and advice and support from Nature Research, please visit <https://www.nature.com/commsenv/article-processing-charges>

At acceptance, you will be provided with instructions for completing this CC BY license on behalf of all authors. This grants us the necessary permissions to publish your paper. Additionally, you will be asked to declare that all required third party permissions have been obtained, and to provide billing information in order to pay the article-processing charge (APC).

Link Redacted

Best regards,

Clare Davis, PhD

Senior Editor

Communications Earth & Environment

www.nature.com/commsenv/

@CommsEarth

REVIEWERS' COMMENTS:

Reviewer #1 (Remarks to the Author):

Changes look good.

Only one minor grammatical correction needed on line 720: change "meeting" to "meet".

Reviewer #2 (Remarks to the Author):

The authors have addressed all my comments.